# The distribution of fitness effects during adaptive walks using a simple genetic network

**Nicholas L. V. O'Brien** [1,2]*, **Barbara Holland** [3,4], **Jan Engelstädter** [1,2], **Daniel Ortiz-Barrientos** [1,2]*

**1** School of the Environment, The University of Queensland, Brisbane, Queensland, Australia, **2** ARC Centre of Excellence for Plant Success in Nature and Agriculture, The University of Queensland, Brisbane, QLD, Australia, **3** School of Natural Sciences, University of Tasmania, Hobart, Tasmania, Australia, **4** ARC Centre of Excellence for Plant Success in Nature and Agriculture, University of Tasmania, Hobart, Tasmania, Australia

☯ These authors contributed equally to this work.
* n.obrien@uq.edu.au (NLVO'B); d.ortizbarrientos@uq.edu.au (DO-B)

**Data Availability Statement:** The data used in this study is available at https://doi.org/10.48610/f3850b0 Instructions and scripts for repeating the analysis and for running the SLiM simulations are available at https://github.com/nobrien97/

## Abstract

The tempo and mode of adaptation depends on the availability of beneficial alleles. Genetic interactions arising from gene networks can restrict this availability. However, the extent to which networks affect adaptation remains largely unknown. Current models of evolution consider additive genotype-phenotype relationships while often ignoring the contribution of gene interactions to phenotypic variance. In this study, we model a quantitative trait as the product of a simple gene regulatory network, the negative autoregulation motif. Using forward-time genetic simulations, we measure adaptive walks towards a phenotypic optimum in both additive and network models. A key expectation from adaptive walk theory is that the distribution of fitness effects of new beneficial mutations is exponential. We found that both models instead harbored distributions with fewer large-effect beneficial alleles than expected. The network model also had a complex and bimodal distribution of fitness effects among all mutations, with a considerable density at deleterious selection coefficients. This behavior is reminiscent of the cost of complexity, where correlations among traits constrain adaptation. Our results suggest that the interactions emerging from genetic networks can generate complex and multimodal distributions of fitness effects.

## Author summary

Historically, models of adaptation have typically considered traits as a sum of effects at many genes. Mutations in these genes move a population incrementally closer to an optimum phenotype. However, the genetic basis of traits is often more complex, with interwoven networks of genes creating non-additive effects that might reduce natural selection's ability to steer populations towards optimal trait combinations. In this study, we developed a model that simulates the evolution of a trait as the product of a gene regulatory network. We used this model to compare the effects of the network on adaptation to a typical additive model. We found that mutations in network populations were more likely to drive the population away from an optimum phenotype. We likened this result to the

NARAdaptiveWalk2023 SLiM modifications are available at https://github.com/nobrien97/SLiM/releases/tag/AdaptiveWalks2023 note that this modification requires the Ascent ODE solution library installed https://github.com/AnyarInc/Ascent/releases/tag/v0.7.0.

**Funding:** This work was supported by an Australian Research Council grant awarded to DO (FT200100169) and the Australian Research Council Centre of Excellence for Plant Success in Nature and Agriculture (CE200100015). The funders had no role in study design, data collection and analysis, decision to publish, or preparation of the manuscript.

**Competing interests:** The authors declare no conflicts of interest.

"cost of complexity", where more complicated genetic systems can create constraints on adaptation. Our results suggest that the non-additive interactions emerging from genetic networks might alter the adaptive dynamics predicted under additive models.

## Introduction

A lingering question in the study of adaptation concerns the distribution of effect sizes of adaptive alleles that affect trait and fitness variation in natural populations. Research across various species and traits has yielded mixed results. In some cases, natural selection favors alleles with small phenotypic effects (e.g. [1–3]), such as the alleles contributing to the evolution of body size in mammals [4]. Other times, however, alleles with larger effects are preferred, such as those found in flower color changes that affect pollinator preferences (e.g. [5–7]). Understanding when allelic effects of different sizes are favored is crucial to comprehend the "adaptive walk"—the metaphorical path a population takes through genetic space as it adapts to its environment [8]. An adaptive walk begins when environmental change induces an "optimum shift", where the phenotype with the highest fitness (the phenotypic optimum) is suddenly changed. This drives the population out of mutation-selection balance and towards the new optimum via directional selection [9]. Each "step" in the walk represents a beneficial allele fixing in the population (i.e. reaching 100% frequency), moving it closer to the new phenotypic optimum [10, 11]. The distribution of fitness effects (DFE) among beneficial alleles is a key feature of adaptation, as it describes the tempo and mode of the adaptive walk in phenotypic space: how many steps there are, how large each step is, and in which direction each step drives the population.

### Adaptive walks and the distribution of fitness effects

The leading theory of adaptive walks was developed by Gillespie and Orr [12–14]. Under this theory, adaptive walks are characterized by a sequence of mutations with diminishing effect sizes. A key prediction of the Gillespie-Orr model is that the fitness effects of beneficial mutations sampled during adaptation form a negative exponential distribution [14, 15]. This shape emerges as populations approach the optimum phenotype: large effect mutations become progressively disadvantageous due to the increased risk of "overshooting" the optimum. Gillespie recognized that extreme-value theory—a statistical branch focused on sampling from extreme tails of distributions—could be integrated into the genetic theory of adaptation to study the DFE. To understand this connection, consider the distribution of fitnesses among all genotypes. Under normal circumstances, the wild-type genotype should have relatively high fitness, and hence it belongs to the right tail of this distribution. The remainder of the right tail of the DFE consists of beneficial mutations. Extreme value theory predicts that this right tail can be described by a family of distributions termed the extreme value distribution (EVD). Gillespie posited that adaptive steps are samples from the EVD [13]. Studying the shape of the EVD highlights the availability of beneficial mutations to a population and which effect sizes are likely to contribute to adaptation.

The EVD can be broadly categorized into one of three "domains of attraction", or shapes: the Gumbel, Weibull, or Fréchet family of distributions (Fig 1; [16–18]). Gillespie argued that the Gumbel is most likely to represent empirical DFEs as it captures common distributions such as the normal, gamma, and log-normal [12, 13]. The Gillespie-Orr exponential expectation outlined above assumes a Gumbel domain [14]. Mutagenesis studies and spontaneous mutation experiments largely support this theoretical expectation (e.g. [14, 19–24]). However,

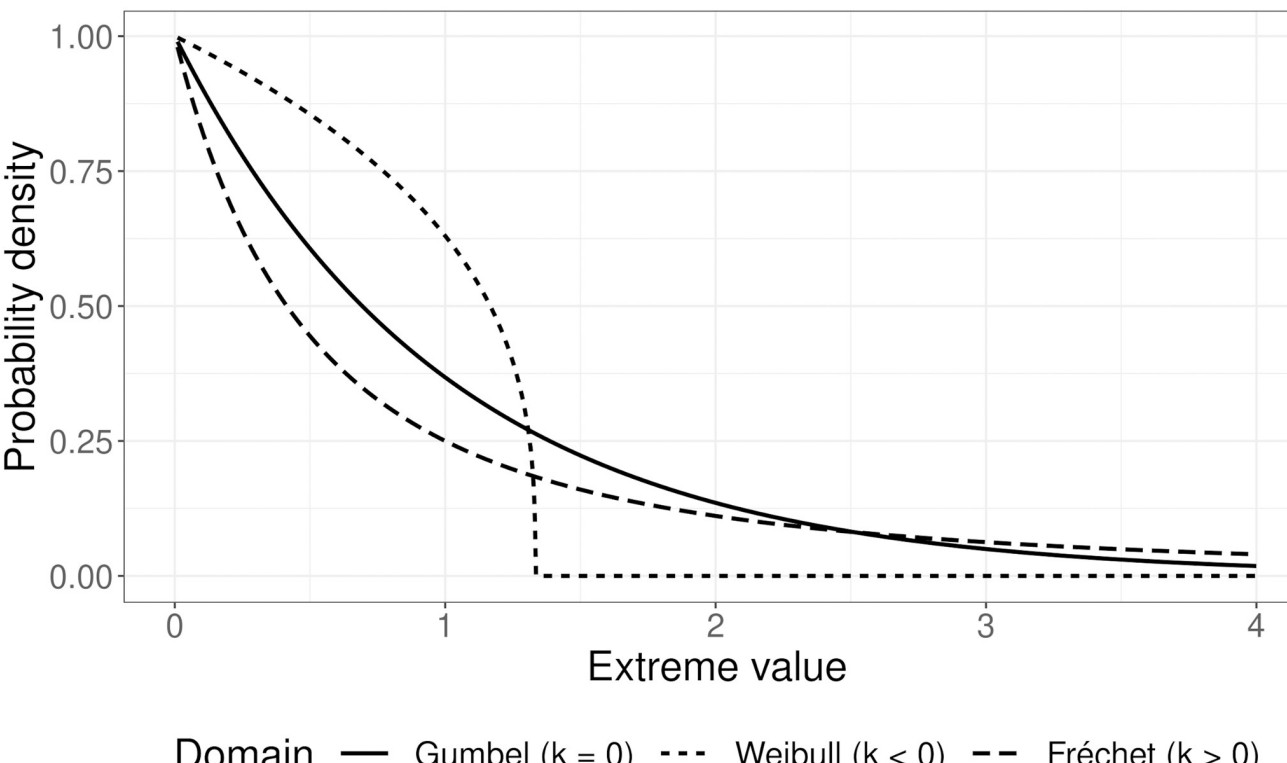

**Fig 1. Examples of how the shape parameter, κ, can change the generalized Pareto distribution (GPD).** The GPD, an extreme value distribution, characterizes the behavior of extreme values: random samples from the right tail of a continuous distribution. Extreme values are used to represent beneficial mutations during an adaptive walk. Depending on κ, the GPD can belong to one of three domains: Gumbel, Weibull, or Fréchet. When κ = 0, the extreme values align with the Gumbel distribution, which is the anticipated distribution for beneficial mutations during an adaptive walk (solid line). If κ < 0, the extreme values take on a Weibull distribution, exhibiting a truncated right tail (dotted line). Decreasing κ shifts the maximum extreme value (the truncation point) towards zero. Conversely, for κ > 0, the extreme values follow the Fréchet distribution, characterized by an extended right tail (dashed line). The specific κ values used in this plot are 0, -0.75, and 1 for Gumbel, Weibull, and Fréchet, respectively.

evidence in favor of alternative domains does exist, suggesting the field is still open to exploration. Adaptive walks with Weibull-distributed DFEs are characterized by fewer large-effect beneficial mutations compared to Gumbel EVDs [25, Fig 1]. Rokyta et al. [16] observed a Weibull distribution in the ID11 ssDNA phage, hinting at an upper limit on the size of beneficial fitness effects. Further, stabilizing selection can generate a Weibull-distributed EVD by limiting the size of beneficial mutations as populations approach the optimum [18]. On the other hand, Fréchet EVDs have more frequent beneficial mutations [25, Fig 1]. Schenk et al. [26] observed a Fréchet EVD in *Escherichia coli* adapting to antibiotics. Although it is unclear as to which conditions cause Fréchet EVDs to arise, environments which invoke strong selective pressures on populations are associated with their appearance [26–29].

There are a large number of factors which can influence the DFE among beneficial mutations and these can produce non-Gumbel EVDs. Some of these factors are environmental, such as stabilizing selection driving Weibull EVDs [18]. However, developmental and selective constraints also contribute to the shape of the DFE, implicating the structure of the genotype-phenotype-fitness (*GPW*) map in the DFE's shape [30–32].

## Genotype-phenotype maps in adaptation

The genotype-phenotype-fitness (*GPW*) map is composed of two distinct components: the genotype-phenotype and the phenotype-fitness relationships. The genotype-phenotype (*GP*)

map describes how developmental and physiological processes translate genotypes into phenotypes, while the phenotype-fitness map describes how ecological and selection regimes create fitness differences between phenotypes. For example, stabilizing selection favors intermediate phenotypes, while directional selection favors extreme phenotypes in one direction [33]. The phenotype-fitness map, along with the related genotype-fitness map, has been extensively studied in quantitative and population genetics (e.g. [34–38]). Because the *GP* map is notoriously challenging to estimate, many quantitative genetics models assume an additive relationship between genotype and phenotype [39, 40]. These models derive from Fisher's [41] infinitesimal model, which supposes that continuous trait distributions can be produced by loci under Mendelian segregation as long as those loci are a) many, and b) have small, additive effects on the phenotype [41]. This model is the basis of modern quantitative genetics, providing a simple *GP* map that requires no information about the underlying genetic systems that describe the developmental and physiological underpinnings of traits [39, 42].

Overwhelming empirical evidence suggests that non-additive gene interactions (epistasis) are ubiquitous in nature [43–45] and play a crucial role in adaptation [46–49]. Additive models, including the infinitesimal, often fail to capture these gene interactions, and when identified, it is unclear how they represent biological gene action [50]. Hence, it is unknown how functional epistasis might contribute to the developmental constraints that limit evolution [44].

Integrating information about the underlying systems describing trait development enables us to capture genetic interactions affecting fitness. Modeling the molecular networks underpinning trait development and expression can help us understand how gene interactions and regulatory processes shape the survival and reproductive success of organisms, and provide a mechanistic view of how variation arises in natural populations. As we delve deeper into the nature of the genotype-phenotype-fitness landscape, it becomes important to re-evaluate our foundational quantitative genetic models in light of our current understanding of the molecular basis of traits. For instance, how does the distribution of fitness effects change when non-additive *GP* maps are considered? Does this affect a population's chance to adapt to a changing environment? To address these questions, we consider a simple gene regulatory network motif to motivate our approach and implement a nonlinear *GP* map into adaptive walk theory.

At the foundation of any trait lies a complex network of interacting genes and regulatory elements which control the expression of proteins [51]. Systems biologists employ mathematical networks known as gene regulatory networks (GRNs) to model such systems [52]. Empirical networks harbor startling complexity: for example, consider the circadian clock network in *Arabidopsis thaliana*. This network is driven by a number of interacting network motifs: small, common subnetworks with particular effects on gene expression. For instance, a feed-forward loop motif in this network generates pulses of expression in *PRR9/7* [53]. In addition, a negative feedback loop (another motif) between *CCA1/LHY* and *PRR5/TOC1* generates a bistable switch [53]. This switch is toggled at daybreak by the aforementioned *PRR9/7* feed-forward loop and again at dusk by yet another circuit [53]. Given the complexity of many GRNs, it is common to study motifs as separate units to elucidate the reasons for their repeated recruitment into so many systems [54–56]. This allows systems biologists to probe the general effects of motifs on a broad range of networks. In this study, we focus on the simplest network motif, negative autoregulation (NAR; Fig 2). This motif consists of two genes: gene *X* activates gene *Z* (indicated by the pointed arrow in Fig 2A), while *Z*'s product limits its further expression (indicated by the flat arrow in Fig 2A).

NAR motifs are highly prevalent in biological networks. Close to half of all transcription factors in *E. coli* are negatively autoregulated [57, 58]. In plants, expression of the oil biosynthesis transcription factor *WRINKLED1* (*WRI1*) is driven via a NAR network that has been

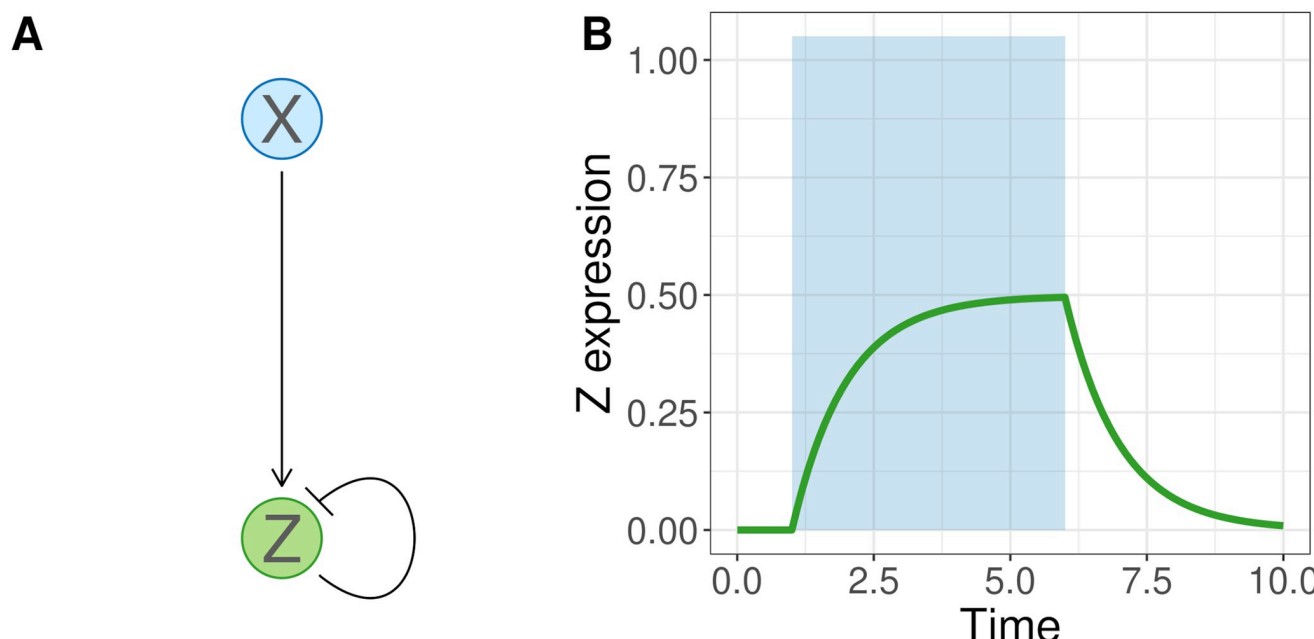

**Fig 2. A negative autoregulation (NAR) motif and its expression curve.** The NAR motif consists of two genes, X and Z. The expression of X activates Z, and the expression of Z inhibits further Z production (A). This results in a characteristic expression curve (B). Z production begins when X is activated (blue shaded area) and stops when it is inactivated. Gene Z product approaches a steady state concentration and then quickly falls off in the absence of X.

evolutionarily conserved from at least the split between monocotyledons and dicotyledons (about 140 million years) [59, 60]. The ubiquity of the NAR motif in nature comes as a result of its self-balancing property: when a gene is overexpressed, the NAR mechanism triggers to reduce that gene's production, leading to a steady state of gene expression [52]. This reduces variability in gene expression between cells and accelerates responses to environmental cues [61, 62]. Owing to its simplicity and ubiquity, we consider the NAR motif a reasonable toy model for beginning to explore the evolution of complex traits mediated by genetic networks.

How might we expect networks to affect adaptation? Genetic networks impose functional epistasis, biological interactions between genes, which can create nonlinear *GP* maps [63]. This results in rugged fitness landscapes where populations can get trapped at local optima because the path towards the global peak (which maximizes fitness) is paved by low-fitness genotypes [34]. Adaptation should then be limited by the structure of the fitness landscape: the more local peaks there are, the less likely it is for a population to be able to find the global optimum. In turn, the structure of the fitness landscape will depend on the nature of the underlying network. For instance, the NAR network provides a relatively simple fitness landscape. Kozuch et al. [64] investigated the fitness landscape of *E. coli*'s *lexA* NAR network. The authors found that the fitness landscape was ridge-like, with fitness maximized along a parameter space that balanced *lexA* production with the strength of autoregulation [64]. However, other motifs might produce different constraints on adaptation. Recent work by Baier et al. [65] found that a synthetic gene network based on a feed-forward motif produced reciprocal sign epistasis, a prerequisite for a rugged fitness landscape. Further, empirical fitness landscapes suggest that while ruggedness is common (although not ubiquitous) [66], connectivity is also high, making these rugged landscapes searchable [67].

In this paper, we use a novel approach to model a quantitative trait as the product of a NAR motif. Previous attempts to reconcile quantitative and population genetics with systems

biology have used a variety of approaches. Some have focused on the explicit modeling of network structures (e.g. [68, 69]), whilst others have considered systems as hierarchical structures of developmental parameters that resemble quantitative traits (e.g. [70, 71]). Our approach is the first to our knowledge to combine both approaches. We model a quantitative trait as the expression of the NAR motif via a system of ordinary differential equations (ODE) similarly to François [69]. However, instead of modeling a gene network's evolvability via the addition/ removal of genes to the network, we instead consider the perturbation of expression dynamics via quantitative changes in the coefficients of the ODE/s, similarly to Slatkin [70]. This approach combines the biological realism of network modeling with the wealth of existing tools for studies of the evolution of quantitative traits. We use our approach to evaluate the shape of the DFE among beneficial mutations during adaptive walks, finding stark differences from classical expectations in some cases. We use Wright-Fisher simulations to describe the adaptive walks of populations following an optimum shift with either an additive or NAR *GP* map, and examine which extreme value distribution domain (Gumbel, Weibull, or Fréchet) best captures the behavior of our model. We then identify if the network imposes constraints on adaptation compared to an additive model. Finally, we discuss the contributions of genetic network architectures to adaptation and propose avenues for further exploration using similar systems models.

## Materials and methods

### The model

**Modeling the NAR motif.**   To model the expression patterns of the NAR motif, we first translate its network diagram (Fig 2A) to a system of ordinary differential equations (ODE). ODEs are commonly used to model gene networks due to their balance between efficiency and realism [72, 73]. The solution to the ODE predicts gene expression over a time period, such as during cell development. The NAR ODE is given by:

$$Z'(t) = \beta_Z \left( \frac{X^h}{K_{XZ}^h + X^h} \right) \left( \frac{K_Z^h}{K_Z^h + Z^h} \right) - \alpha_Z Z \tag{1}$$

The coefficients in this equation have biological relevance. $X$ and $Z$ represent the cellular concentrations of the two genes. $h$ is the Hill coefficient reflecting the sensitivity of the system to the presence of $X$ and/or $Z$ products. Higher values indicate a more rapid response to increasing $X$ and/or $Z$ concentration [52]. $K_{XZ}$ and $K_Z$ are activation and repression coefficients, respectively. They control how quickly the presence of $X$ or $Z$ drive the activation or suppression of further $Z$ expression [52]. In this study, we fixed these parameters (values can be found in S1 Table) in favor of studying the evolution of the remaining two: $\alpha_Z$ and $\beta_Z$.

$\alpha_Z$ is the rate at which $Z$ is removed from the cell. This might reflect, for example, the activity of ubiquitinase in tagging $Z$ for removal and the 26S proteasome in breaking down the protein [74, 75]. $\beta_Z$ represents the production rate of $Z$, which is influenced by factors such as transcription factor binding affinity, enhancers, silencers, and *trans*-acting regulatory elements [76]. These biological interpretations provide some realism to modeling quantitative traits. It also gives interpretability to the action of mutations that affect these coefficients, which will be explained in the sections following.

$X$ follows a step function and is only expressed within the interval $t \in [t_{\text{start}}, t_{\text{stop}}]$:

$$X(t) = \begin{cases} 1, & \text{if } t_{\text{start}} \leq t \leq t_{\text{stop}}, \\ 0, & \text{otherwise} \end{cases} \tag{2}$$

where $t_{\text{start}}$ and $t_{\text{stop}}$ are the time points at which $X$ is activated and deactivated. Values for these parameters are given in S1 Table.

The solution of the ODE is characterized by a nonlinear approach to maximum $Z$ expression, followed by a decline after $X$ expression ceases (Fig 2B). The area underneath the expression curve (Fig 2B) is the total amount of $Z$ produced, which we can take as a quantitative trait value. To create variation in this trait value, the coefficients of Eq 1 can be varied. We refer to these coefficients as "molecular components". In this study, we focus on modeling mutations in two of the NAR's molecular components: the $Z$ degradation rate, $\alpha_Z$, and the $Z$ production rate, $\beta_Z$. Mutations in our model have direct effects on either $\alpha_Z$ or $\beta_Z$. This leads to a differently-shaped expression curve when the ODE is solved, and hence a (potentially) different trait value.

We adopt a model where loci contribute multiplicatively to the values of molecular components. We refer to these loci as molecular quantitative trait loci (mQTLs). The multiplicative transformation ensures that the molecular components are always positive, which is essential as these values represent rates which cannot be negative. We study the evolution of the mQTLs using the Wright-Fisher model, a foundational model in population genetics which describes the stochastic process of allele frequency change in a finite population [34, 77]. In our implementation, individuals in the Wright-Fisher population are diploid.

Although this implementation is complex for a simple adaptive walk scenario, explicitly modeling the entire population means that the adaptive walks can emerge organically and situations with small deviations from strict adaptive walk scenarios (where sometimes more than two alleles segregate in a population) are also covered. Also, this means the model is easily extensible to many different genetic architectures with different networks, numbers of contributing loci, recombination rates, and mutational effect sizes. For a more abstract formulation of such a generalised model, refer to S1 Appendix.

**Phenotype calculation.** To calculate an individual's phenotype from their set of mQTLs, we follow the below algorithm:

1. Take the exponent of the sum of all alleles for $\alpha_Z$ across all loci and both homologous chromosomes (this is analogous to an additive model summing effects across all loci, but with a multiplicative transformation as described above).

2. Repeat step 1 for $\beta_Z$.

3. Substitute the $\alpha_Z$ and $\beta_Z$ values into the ODE, Eq 1, and solve the ODE between time points 0 and 10 to get an expression curve.

4. Take the area under the expression curve to get the total amount of $Z$ expression (the phenotype).

5. Repeat steps 1–4 for each individual in the population.

This algorithm is described mathematically below.

Let **C** be a vector of the molecular components:

$$\mathbf{C} = (\alpha_Z, \beta_Z)$$

An allele at locus $i$ and chromosome $j$ has an effect $a^{ij}$ on one of the molecular components.

Given an individual's alleles, the value for each molecular component is calculated by exponentiation of the sum of allelic effects across all mQTLs and chromosomes. For molecular

component $k$:

$$C_k = \exp\left(\sum_{i=1}^{L_Q}\sum_{j=1}^{2} a_k^{ij}\right),\tag{3}$$

where $C_k \in [0, \infty)$. $L_Q$ represents the number of causal loci along the genome. In all simulations, $L_Q = 2$. This treatment of allelic effects assumes no explicit epistasis or dominance deviations, however these can arise as consequences of the nonlinear ODE solution.

After calculating the molecular component values (**C**), we can determine the amount of $Z$ expression, the phenotype. We achieve this by constructing a system of ODEs (Eq 1) and solving for the area under the expression curve (i.e. the total $Z$ produced during a time period).

We express the ODE as an initial value problem where given the "initial" or starting value of the function we can determine the function's behavior for subsequent time points using the ODE:

$$\begin{aligned} Z'(t) &= F(Z(t)) \\ Z(0) &= Z_0 \end{aligned}$$

where $F$ is the function that defines the differential equation. To calculate $Z$, we integrate the expression curve:

$$Z = \int_0^{t_{\max}} Z(t)dt\tag{4}$$

where $t_{\max} = 10$.

In Eq 2, $t_{\text{start}} = 1$ and $t_{\text{stop}} = 6$. $Z$ activity is evaluated for $0 \leq t \leq 10$. We chose $t_{\max} = 10$ to allow for a wide range of possible phenotypes while limiting the computational cost of solving the ODE. $t_{\text{start}}$ and $t_{\text{stop}}$ were chosen so that $X$ is activated half of the total time evaluated.

**Fitness calculation.** To map the trait value ($P$) to fitness ($w$), we utilize a Gaussian fitness function, which allows us to model directional and stabilizing selection. The Gaussian fitness function, originally described by Lande [40], is defined as follows:

$$w(P) = \exp\left(\frac{-\Delta z^2}{2\sigma^2}\right)\tag{5}$$

In this function, $\Delta z = P - P_O$, where $P_O$ is the optimal phenotype, and $\sigma$ is the width of the fitness function. Wider functions represent weaker selection.

**Evolutionary simulation.** To model the evolution of the network over multiple generations, we employ a Wright-Fisher (WF) model [34, 78]. We consider a diploid population of $N = 5000$ individuals with random mating and non-overlapping generations. This satisfies the requirement that $2N\mu \ll 1$ (where $2N$ is the total number of genomes in the population; $2N\mu = 0.0915$), which is necessary to meet the Gillespie-Orr model's assumption of strong selection relative to mutation (see section Model validation for more information on this) [12, 79]. Offspring are generated by sampling with replacement from parents, with the sampling probability weighted by their relative fitness. Individuals possess a pair of homologous chromosomes with two mQTLs: one for each molecular component. Random mutations occur at a rate of $\mu = 9.1528 \times 10^{-6}$ per locus per generation, for a total rate of $1.831 \times 10^{-5}$ across both loci. This mutation rate is based on the average mutation rates observed in *A. thaliana* and adjusted to per-locus rates by multiplying it with the average length of a eukaryotic gene (1346bp) [80, 81]. *A. thaliana* was chosen as it is a model plant species with good estimates of mutation rates. The mutational effects of mQTLs on molecular components were drawn from a standard

normal distribution and the effect added to the previous allele at that locus such that $a_{new} \sim \mathcal{N}(a_{old}, 1)$, where $a_{new}$ is the new allele at a given locus and $a_{old}$ is the previous allelic effect at that locus. After being sampled, the allelic effects were exponentiated as per Eq 3. We assume free recombination between loci.

We also considered an additive model. In this scenario, genomes also consisted of two causal loci to match the mutational target size of the NAR model. However, the trait value was given by summing allelic effects across QTLs instead of solving an ODE:

$$P_{\text{Additive}} = \sum_{i=1}^{L_Q} \sum_{j=1}^{2} a_k^{ij}, \tag{6}$$

where $a_k^{ij}$ is the allelic effect of locus $i$ on chromosome $j$.

## Computational implementation

To investigate our model's behavior during an adaptive walk, we implemented the model in a custom version of the forward-time, individual-based simulation software SLiM 3.7.1 [82]. Our modified SLiM implementation can be accessed openly at https://github.com/nobrien97/SLiM/releases/tag/AdaptiveWalks2023. SLiM scripts are available at https://github.com/nobrien97/NARAdaptiveWalk2023. A flowchart of the SLiM implementation is given in S1 Fig. The ODE was solved by integrating the Ascent numerical solution library [83] into SLiM. Ascent solved Eq 4 to produce trait values for the individuals in the population. Since individuals often shared genotypes (and hence had the same ODE inputs), ODE solutions were cached in memory to reduce redundant solutions and improve performance.

Both models underwent 50,000 generations of burn-in. The burn-in ensured that populations had high fitness just prior to the optimum shift, an assumption of the Gillespie-Orr model [12–14]. During burn-in, populations adapted to a phenotypic optimum at $P_O = 1$. The phenotypic optimum was then instantaneously shifted to $P_O = 2$, and the population was monitored for 10,000 generations of adaptation. We measured the phenotypic means and allelic effects of segregating and fixed mutations every 50 generations. We replicated each model 2,880 times with 32-bit integer seeds sampled from a uniform distribution in R 4.3.1 [84]. Detailed simulation parameters can be found in Table 1. Simulations were executed on the National Computational Infrastructure's Gadi HPC system.

**Table 1. Simulation parameters.** Table of symbols, names, descriptions, and values for relevant parameters used in the forward-time Wright-Fisher simulations.

| Symbol | Parameter | Description | Value |
|---|---|---|---|
| $N$ | Population size | Number of individuals in the population | 5,000 |
| $\mu$ | Mutation rate | Per-locus per-generation rate of mutation in the genome | 9.1528e-06 |
| $\tau$ | Mutational effect variance | Variance in the sampling distribution of mutational effects on molecular components, $a \sim \mathcal{N}(0, \tau)$ | 1.0 |
| $\sigma$ | Fitness function width | Width of the Gaussian fitness function (5% drop in relative fitness for individuals one unit away from the optimum). | $\sqrt{10}$ |
| $L_Q$ | Number of loci | The number of causal loci contributing to the phenotypic trait/molecular components | 2 |
| $O_{\text{shift}}$ | Optimum shift amount | Magnitude of shift in the phenotypic optimum after the burn-in phase | 1 |
| $n_{\text{burn-in}}$ | Number of burn-in generations | Number of generations of burn-in prior to shifting the optimum. | 50,000 |
| $n_{\text{gen}}$ | Number of adaptation generations | Number of generations the model was run for following the optimum shift | 10,000 |

Of the simulation parameters, the most important were the fitness function width, $\sigma$, and the optimum shift amount, $O_{shift}$. The Gillespie-Orr model assumes that the wild-type has relatively high fitness, so that in the total space of genotypes, it exists on the right-tail of the distribution [14]. Hence, $\sigma$ and $O_{shift}$ need to be chosen so that at the optimum shift (generation 50,000) fitness is still relatively high. We chose $\sigma$ and $O_{shift}$ so that this assumption is met: at the optimum shift, individuals perfectly adapted to the burn-in optimum suffered a $\sim 5\%$ drop in relative fitness.

## Model validation

The Gillespie-Orr model assumes that there is strong per-locus selection relative to mutation [12, 13]. This regime is often referred to as the strong selection weak mutation (SSWM) paradigm, as opposed to the weak selection strong mutation regimes found in polygenic models (e.g. the infinitesimal model) [41, 85].

To ensure we were in the SSWM domain assumed by adaptive walk theory, we measured population heterozygosity and the effects of segregating alleles on trait variation. To measure heterozygosity, $H$, we used the equation

$$H^i = N_{het}^i/N \tag{7}$$

where $N_{het}$ is the number of individuals heterozygous at locus $i$, and $N$ is the total population size. Since we had two genes in our simulations (one for each molecular component, and two for the additive model), we took the average heterozygosity across both. Under SSWM theory, heterozygosity should be close to zero, as evolution is mutation limited and beneficial alleles should be quickly fixed. To measure the effects of segregating alleles on the trait, we measured the ratio between the phenotype created by only fixations and the population mean phenotype,

$$r = P_{fixed}/\bar{P} \tag{8}$$

where $P_{fixed}$ is the phenotype due to only fixed alleles and $\bar{P}$ is the mean population phenotype. $P_{fixed}$ was calculated by removing any segregating effects from the $\alpha_Z$ and $\beta_Z$ values and recalculating the phenotype via Eq 4. Under SSWM, $r \approx 1$, as the phenotype of the population should be constructed from only fixations due to strong selection quickly fixing any segregating beneficial alleles.

## Fitness effect calculation

To assess the shape of the DFE, we needed to measure the fitness effects of mutations that arose during the adaptive walk. We performed single gene knockouts and measured the difference between the "wild-type" (genotype AA) and knockout's (genotype aa) relative fitness. A flowchart is given in S2(A) Fig. To measure a given mutation's effect on fitness, we first subtracted that mutation's homozygous effect on the molecular component ($\alpha_Z$ or $\beta_Z$) and recalculated the trait value via Eq 4. We then calculated the relative fitness of this knocked-out individual ($w_{aa}$) via Eq 5. To get our final homozygous fitness effect, we subtracted $w_{aa}$ from the relative fitness of an individual with the mutation ($w_{AA}$):

$$s = w_{AA} - w_{aa} \tag{9}$$

We built a custom tool to recalculate the phenotype and fitness recalculations, implementing the Ascent C++ library to solve ODEs [83]. This library was also used in the SLiM

simulations, keeping the phenotype calculation consistent. The source code for our tool can be found at https://github.com/nobrien97/odeLandscape. Fitness effects were calculated in a similar manner for additive simulations. However, the phenotype was computed as the sum of additive effects instead of using the network ODE solution.

### *In silico* mutant screen

In addition to the DFE of mutations that arose during the adaptive walk, we were also curious about the total DFE across the space of possible mutations (including deleterious mutations that would be lost quickly). We ran an *in silico* mutant screen experiment to determine whether the fitness distribution of possible mutations differed between additive and NAR populations. A flowchart is given in S2(B) Fig. We sampled 1,000 mutations from a standard normal distribution and added them to the molecular component values/quantitative trait values of individuals at each step of the adaptive walk. This was done for both network and additive models. We chose a standard normal distribution as it matched the distribution of new mutations that populations faced in the simulations. For the network models, mutations were applied separately to $\alpha_Z$ and $\beta_Z$, while in the additive models, the mutation effect was added to the quantitative trait value. We then recalculated the phenotypic and homozygous fitness effects of the sampled mutations using the fitness effect calculation methods described above. The source code for this experiment can be found at https://github.com/nobrien97/NARAdaptiveWalk2023.

### Expected waiting times to beneficial mutation

To tie the mutant screen findings to a quantitative measure of adaptation, we used the mutant screen fitness effect distribution to estimate the expected waiting time to a beneficial mutation between models. We first calculated waiting times, $t_{\text{wait}}$ for a particular simulation replicate at a given adaptive step using the equation

$$t_{\text{wait}} = \frac{1}{4N\mu p_{s>0}} \tag{10}$$

where $N$ is the population size, $\mu$ is the per-locus, per-generation mutation rate, and $p_{s>0}$ is the probability that a new mutation is beneficial. $N$ is multiplied by 4 because the population is diploid, and because there were two loci contributing to the trait. $p_{s>0}$ is given by

$$p_{s>0} \approx \frac{m_{s>0}}{m}$$

where $m$ is the total number of mutations generated in the *in silico* mutation screen experiment and $m_{s>0}$ is the number of mutation screen alleles with beneficial effects on fitness.

Waiting times were calculated for each adaptive step and model. Comparisons between models were done using a bootstrap analysis. We sampled 100,000 different NAR-Additive model pairs across all adaptive steps, calculating the difference between their waiting times. We then calculated the means and 95% confidence intervals (CIs) for each model's waiting times and the difference between models ($\Delta t_{\text{wait}}$). The source code for this analysis can be found at https://github.com/nobrien97/NARAdaptiveWalk2023. A flowchart describing this methodology is shown in S3 Fig.

We also measured the difference in $p_{s>0}$ between models and across adaptive steps using a linear model:

$$
\begin{aligned}
p_{s>0} = \quad & \alpha + \beta_1(\text{Model}_{\text{NAR}}) + \beta_2(\text{Adaptive step}_1) + \\
& \beta_3(\text{Adaptive step}_2) + \beta_4(\text{Adaptive step}_{\geq 3}) + \\
& \beta_5(\text{Model}_{\text{NAR}} \times \text{Adaptive step}_1) + \\
& \beta_6(\text{Model}_{\text{NAR}} \times \text{Adaptive step}_2) + \\
& \beta_7(\text{Model}_{\text{NAR}} \times \text{Adaptive step}_{\geq 3}) + \epsilon
\end{aligned}
\tag{11}
$$

Where $\alpha$ is the intercept, $\beta_i$ is the slope coefficient for dependent variable $i$, and $\epsilon$ is the residual error. To estimate marginal means and contrasts, we used the R package `emmeans` 1.8.5 in R 4.3.1 [84, 86].

## Fitness landscapes

With the mutant DFE evaluated, we turned to how the underlying network might drive the complex distribution that we discovered. This involved constructing a fitness landscape to show how fitness changed with different combinations of $\alpha_Z$ and $\beta_Z$, introducing epistasis to the system. We generated 160,000 combinations of $\alpha_Z$ and $\beta_Z$, sampling $\alpha_Z$ and $\beta_Z$ values between 0 and 3. We then calculated the phenotype and fitness for each combination using the ODE landscaper tool we used to calculate fitness effects. We plotted the resulting fitnesses against $\alpha_Z$ and $\beta_Z$ using `ggplot2` 3.4.2 [87] in R 4.3.1 [84]. A similar method was used to estimate the fitness landscape of the ratio $\beta_Z/\alpha_Z$ as $\alpha_Z$ increased. For this analysis, we generated another 160,000 combinations of $\alpha_Z$ and $\beta_Z/\alpha_Z$ ratios. Both $\alpha_Z$ and $\beta_Z/\alpha_Z$ values were sampled between 0 and 3.

A key point to note is that the DFE and the fitness landscape are reflections of each other. The fitness landscape represents the entire space of possible $\alpha_Z$ and $\beta_Z$ combinations, whereas the DFE that we estimated during the mutation screen experiments is a subspace of that landscape. This subspace is the area that a population can reach via a single mutational step. Hence, the DFE represents the explorable portion of the fitness landscape for a population at a particular point in time, whereas the fitness landscape is the total space explorable by any population at any time, given a specific genetic architecture and phenotypic optimum. Neither depend on the evolutionary history of the population nor the SSWM assumption of this study.

## DFE analysis

The final part of our analysis involved measuring the shape of the distribution of beneficial fitness effects (DFE among beneficial mutations), which should be Gumbel-distributed under Gillespie-Orr predictions [13, 15]. We used a method developed by Beisel et al. [88] to fit a generalized Pareto distribution (GPD) to the DFE among beneficial mutations obtained from the mutant screen. A flowchart describing this approach is provided (S4 Fig). The GPD characterizes a distribution of extreme events that exceed a threshold. The threshold is the high-fitness wild-type and the extreme events are beneficial mutations. The shape parameter of the GPD, $\kappa$, indicates which domain of attraction the underlying distribution belongs to. Beisel et al.'s [88] method uses a likelihood ratio test to evaluate if $\kappa = 0$, corresponding to the Gumbel case. If $\kappa > 0$, the DFE belongs to the Fréchet domain, and if $\kappa < 0$, it belongs to the Weibull domain [88, 89].

We used two different sampling approaches to fit our data to GPDs (S4 Fig). In the first method, we pooled mutant screen alleles across all replicates to create a pooled DFE of beneficial mutations for additive and network models. We sampled 1,000 mutations from the pooled DFE and fit the GPD to the resulting sampling distribution. We repeated this process 10,000 times, yielding a distribution of estimates of $\kappa$. The pooled approach shows the average shape of the DFE among beneficial mutations across many adaptive walks. In the second method, we sampled up to 100 mutations from each simulation's mutant screen DFE of beneficial mutations and fit the GPD to that sample. This was done per adaptive step. Simulation-step pairs with fewer than 20 beneficial alleles were discarded as they could not be reliably used to estimate the GPD shape [88]. We obtained a distribution of $\kappa$ estimates, one for each simulation-step pair. This approach shows the variability in the shape of the DFE during different adaptive walks. In both methods, p-values from the likelihood ratio tests were combined across replicates using Fisher's method [90]. Both approaches fit the GPD using the R package `GenSA` 1.1.8 [91] in R 4.3.1 [84], with code modified from Lebeuf-Taylor et al. [92].

## Results

### Assumption validation

We first checked if we were in the strong selection weak mutation (SSWM) regime assumed by the Gillespie-Orr model. Under SSWM, each beneficial mutation should fix before another arises. Hence, heterozygosity at QTLs/mQTLs should be close to zero most of the time. We found observed heterozygosity ($H$) at the QTLs/mQTLs was low on average, peaking just after the optimum shift at 11.39% ± 0.423% (95% confidence interval, CI) and 9.637% ± 0.400% (95% CI) in additive and NAR models, respectively (S8 Fig). The grand mean $H$ across all time points was 8.368% ± 0.028% (95% CI) and 7.841% ± 0.027% (95% CI) for additive and NAR models. Although $H$ was not zero, the majority of variation arising was not beneficial (Fig 5B), meaning that the chance that two beneficial mutations could co-segregate was lower than $H$ suggests.

To further test the SSWM assumption, we also measured the ratio between phenotypes generated by only fixations (i.e. the phenotype in absence of all segregating variants) and the mean population phenotype (Eq 8. This ratio describes how much trait variation was contributed by segregating variants compared to fixations. When $r = 1$, segregating variants contribute no trait variation. This is the SSWM expectation. When $r > 1$, segregating variants decrease the trait value on average compared to the phenotype due to fixations, and vice versa when $r < 1$. We found that most populations had no variation contributed by segregating variants (S9 Fig). The average $r$ was 1.019 ± 0.007 (95% CI) and 1.024 ± 0.011 (95% CI) for additive and NAR populations respectively. From these figures, it seems that the populations are in the SSWM domain, enabling us to explore the extreme value theory expectations set out by Gillespie and Orr [12–14].

### NAR and additive populations differ in their ability to adapt

We first examined the phenotypic response to selection during adaptive walks in populations with either an additive or a network (NAR motif) *GP* map. We defined an adapted population as one with a mean phenotype within 10% of the optimum ($1.9 < \bar{P} < 2.1$). We chose 10% as the threshold as it only decreased fitness by 0.04% relative to a 5% threshold, but increased the sample size of adapted populations by 42% (2636 adapted populations vs. 1529). By the end of the simulation, 50.5% of the additive simulations and 41% of the network simulations had adapted. For the remainder of the results, we consider only the adapted populations.

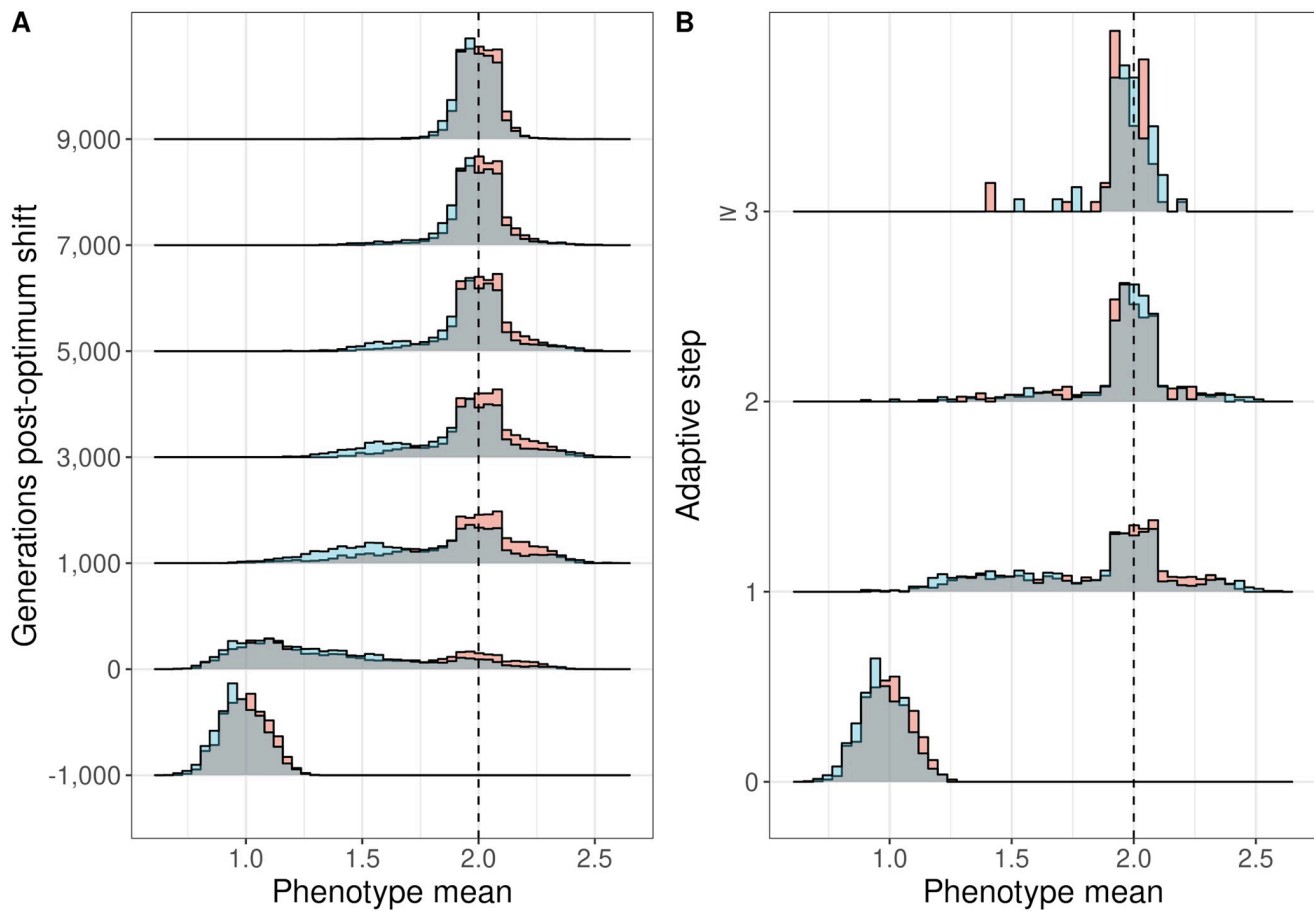

**Fig 3. Phenotypic evolution over 10,000 generations of adaptation across 2,880 replicates of additive and network models.** (A) The distribution of mean phenotypes among replicate populations during the adaptive period. The -1000 timepoint represents 1,000 generations before the optimum shifted. (B) The mean phenotype at each adaptive step (each fixation along the adaptive walk). The last step is pooled across all steps greater than or equal to step three in the walk. The dotted line in A and B represents the post-shift optimum. The grey area shows the overlap between additive and network distributions.

Among the adapted populations, there were differences in the rate of trait evolution between the models (Fig 3). Network model populations took longer to reach the optimum on average (Fig 3A), although the changes in phenotype across adaptive steps were similar between the models (Fig 3B). Note that this difference cannot be solely attributed to the NAR ODE, as the multiplicative scaling of the allelic effects also contributes to differences between the models. See the Discussion for more on this distinction. There was no difference between the models in the number of adaptive steps taken. Of the populations that adapted, 66.3% did so in a single step, 28% in two, and the remaining 5.7% adapted in three or more steps. On average, the number of steps taken was 1.4 ± 0.026 (95% CI). This agrees with results from Orr [93], who found a mean walk of $\approx 1.72$ steps, assuming that the genotype was comprised of many loci and that the starting genotype of the walk was randomly sampled.

Among populations that had not yet adapted after one step (the left tail in Fig 3), the median step in a network model occurred 900 (95% CI [150, 1600]) generations later than in the additive models (S5 Fig). To investigate the underlying causes of the differences in adaptation success between models, we examined the distribution of beneficial fitness effects among fixations during the adaptive walk.

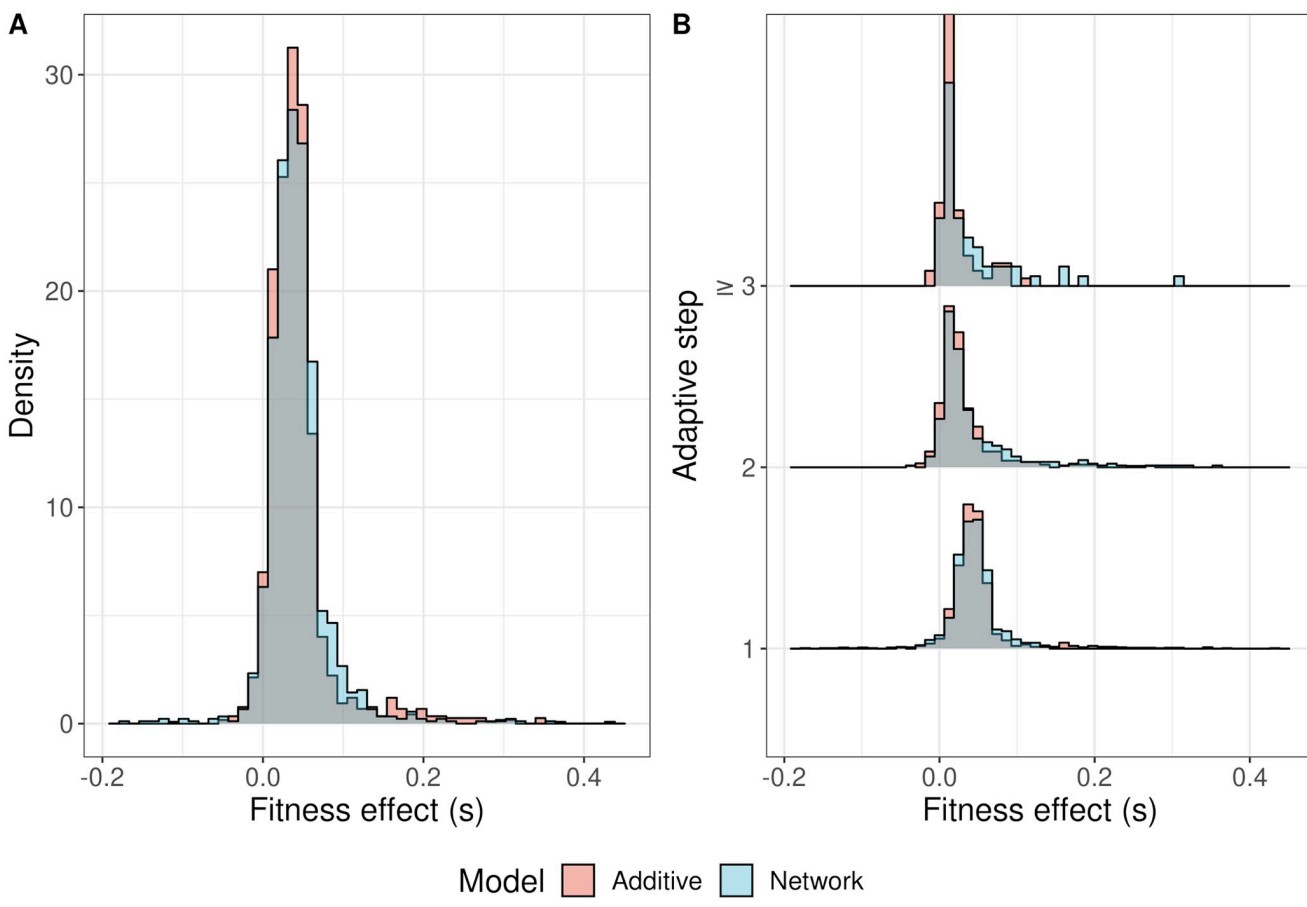

**Fig 4. Fitness effects among fixations during adaptive walks to a phenotypic optimum across 2,880 replicates of additive and network models.** (A) The overall distribution of all fixations across the entire walk. (B) The distribution of fixations at each adaptive step. The grey area shows overlap between the additive and network distributions of effects.

### The distribution of beneficial fitness effects among fixations is similar between NAR and additive populations

Fixations had similar effect sizes in both models and became smaller over the course of the adaptive walk (Fig 4). We found that the mode of the distribution of beneficial fixations was offset from 0 (S6 Fig), supporting a gamma distribution of fixations rather than an exponential distribution. We fit gamma distributions to the DFE among fixations in both the additive and network models using `fitdistrplus` 1.1.8 in R 4.3.1 [84, 94]. The deleterious fixations shown in Fig 4 were excluded from the dataset when fitting the gamma distribution. These deleterious fixations were neutral or adaptive prior to the shift in the optimum and typically reached high frequencies before the adaptive walk commenced (S7 Fig). Therefore, these fixations can be attributed to the action of genetic drift.

We compared the shape and rate parameters of the gamma fits between the models. The network model populations exhibited slightly fewer small effect fixations and had a shorter tail compared to the additive models (NAR: Shape = 2.032 ± 0.147 95% CI, Rate = 44.392 ± 3.651 95% CI; Additive: Shape = 1.722 ± 0.108 95% CI, Rate = 37.588 ± 2.730 95% CI). However, the DFE among beneficial fixations did not differ strongly between the models. To investigate this further, we needed to determine the availability of beneficial mutations to the populations. We

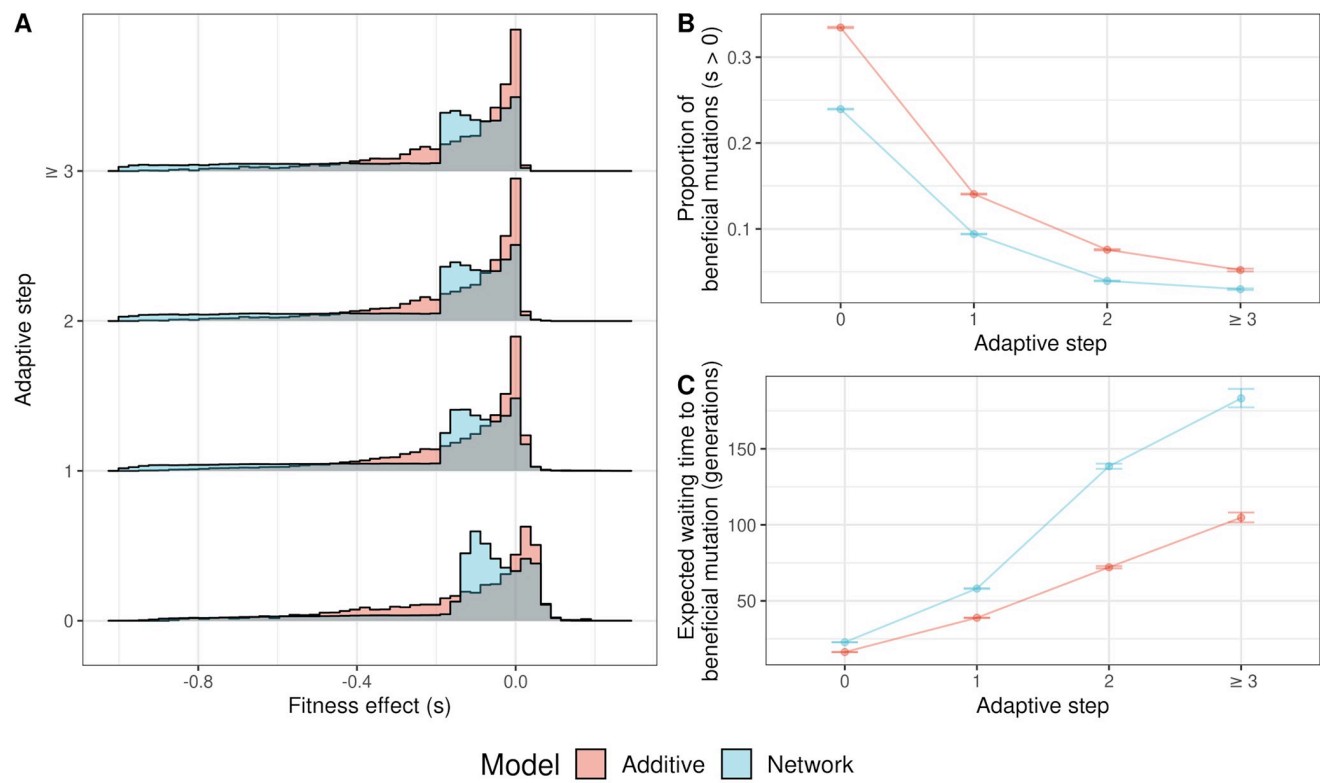

**Fig 5. Fitness effects among possible mutations and the consequences for adaptation in 2,880 replicates of additive and network models.** (A) The distribution of fitness effects among 1,000 randomly sampled mutations in additive and network models. The grey area represents overlap between additive and network models. (B) The proportion of mutations that are beneficial over the adaptive walk. (C) The change in beneficial mutation probability corresponds to the expected waiting time for a beneficial mutation.

did this by conducting an *in silico* mutant screen experiment, examining the DFE among new mutations at each adaptive step.

## The distribution of fitness effects among possible mutants is complex and deleterious in NAR populations

The *in silico* mutation screen experiment revealed that the DFE among new mutations differed dramatically between models (Fig 5A). Both were strongly negatively skewed; however, mutations in NAR populations were overall more deleterious than those in additive models, with a longer and more pronounced tail of strongly deleterious mutations. In addition, the DFE in NAR populations was bimodal (Fig 5A). The first mode was at $s \approx 0$, similar to the additive DFE mode (NAR: $s = -0.0003$; Additive: $s = -0.0004$). The second mode was at $s = -0.113$. Among beneficial mutations, the distributions were more similar.

## Data pooling affects generalized Pareto distribution fitting results

We fit a generalized Pareto distribution (GPD) to the DFE among beneficial mutations generated by the mutant screen experiment using methods developed by Beisel et al. [88]. This was done separately for additive and network models. The shape parameter of the GPD ($\kappa$) describes what domain the DFE belongs to: $\kappa = 0$ specifies a Gumbel domain, $\kappa < 0$ specifies a Weibull domain, and $\kappa > 0$ specifies a Fréchet domain. We fit this in two ways: first, by pooling all replicates and sampling from this pooled distribution to estimate the GPD parameters;

and second, by sampling each replicate at each of its adaptive steps separately and estimating the GPD for each estimate (for more information, see the Methods section). Under the pooled approach, we found evidence for a minor deviation from an exponential distribution towards the Weibull domain in both the additive and network models (Network: $\hat{\kappa} = -0.130 \pm 0.0007$; Additive: $\hat{\kappa} = -0.118 \pm 0.0005$). However, this deviation is small enough that the adaptive walk should behave as if its DFE belonged to the Gumbel domain [17].

Under the per-simulation method, we saw a different result. The DFE among beneficial mutations in both NAR and additive populations were strongly Weibull distributed. In the NAR model, $\kappa$ tended to decrease over the adaptive walk. However, the additive model $\kappa$ estimate was relatively stable at around $\kappa \approx -2$, slightly increasing over the adaptive walk (S11 Fig and S2 Table). In addition, these non-pooled fits were considerably more variable than under the pooled approach (S10 Fig), particularly at early adaptive steps. Under the per-simulation/adaptive-step approach, across all adaptive steps, 99.9% of additive replicates and at least 99.6% of NAR replicates had DFEs of beneficial mutations that fit the Weibull domain. Under the pooling method, all replicates in both models were approximately Gumbel-distributed (i.e. $\kappa \approx 0$).

## Beneficial mutations are less common in NAR populations than additive populations

Despite sharing similar DFEs, beneficial mutations were on average less common in network models than in additive models (Linear model, Eq 11; $F_{7,5171} = 1239$, $R^2 = 0.626$, Fig 5B). The difference between models diminished with adaptive steps (as populations neared the optimum). Immediately after the optimum shift, mutations in NAR populations were 9.504 ± 0.639% (95% CI) less likely to be beneficial. At the first adaptive step, this difference decreased to 4.648 ± 0.639% (95% CI). At the second adaptive step, the NAR models were 3.630 ± 1.107% (95% CI) less likely to produce a beneficial mutation. At further adaptive steps, there was no difference between models. We measured the effect of the lower beneficial mutation rate on adaptation by estimating the waiting time for a new beneficial mutation to arise. We found that the mean expected waiting time for a beneficial mutation in NAR populations was 98.555 ± 5.816 (95% CI) generations longer than in additive populations (Additive mean waiting time: 136.382 ± 2.131 (95% CI) generations; NAR mean waiting time: 234.937 ± 5.414 generations; Fig 5C). Given that the DFE differed between models, we wanted to explore the causes underpinning this. To do so, we investigated the fitness landscape of $\alpha_Z$ and $\beta_Z$.

## The NAR fitness landscape is ridge-like across molecular component space

We found a pronounced fitness ridge on the NAR fitness landscape, surrounded by valleys of low fitness (Fig 6A). This ridge lay diagonally across $\alpha_Z$ and $\beta_Z$ space, suggesting that fitness depended on the ratio of the molecular components. This relates to the underlying ODE: $\beta_Z/\alpha_Z$ is the level of steady state $Z$ expression under simple gene regulation (i.e. without any negative feedback or induction) [52]. Plotting this ratio over $\alpha_Z$ confirmed this: $\beta_Z/\alpha_Z$ was the arbiter of fitness as long as $\alpha_Z \gtrapprox 0.5$ (Fig 6B). We investigated how the $\beta_Z/\alpha_Z$ ratio contributed to the shape of the *GP* map, finding a nonlinear relationship between $\beta_Z/\alpha_Z$ and phenotype (Fig 6C). Around the optimum, increasing $\beta_Z/\alpha_Z$ quickly increases the trait value, however with larger $\beta_Z/\alpha_Z$ increases, the trait value changes less (Fig 6C). We approximated the optimum $\beta_Z/\alpha_Z$ ratio ($\theta_{\beta_Z/\alpha_Z}$), by substituting our fixed molecular components ($K_{XZ} = 1$, $K_Z = 1$, $h = 8$)

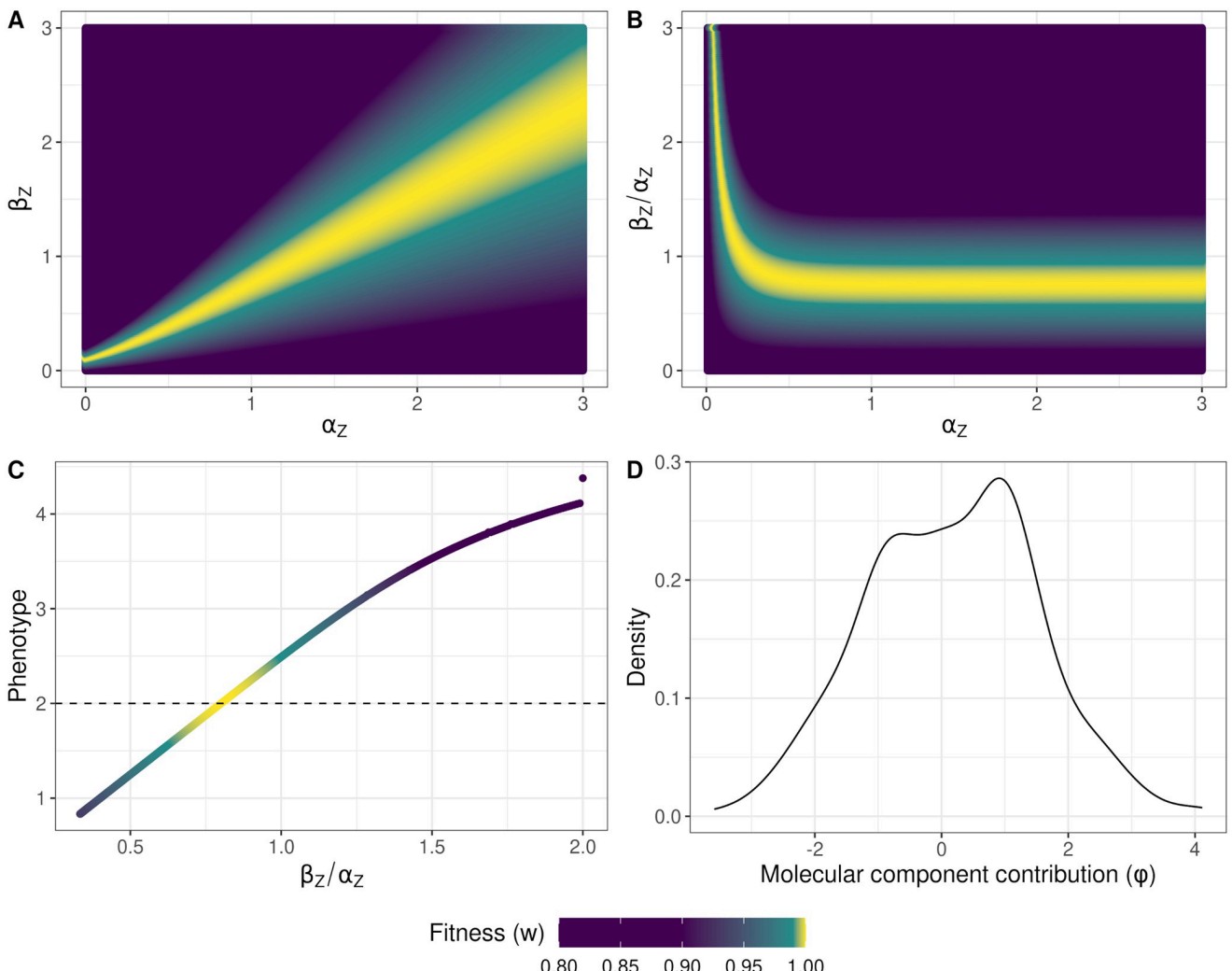

**Fig 6. The fitness and phenotype landscapes of the network model.** (A) The fitness landscape with regards to the molecular components of the network, $\alpha_Z$ and $\beta_Z$. The landscape consists of a high-fitness ridge (yellow) surrounded by low-fitness valleys (purple). (B) The ratio of $\beta_Z/\alpha_Z$, the steady state of the network, is what matters for adaptation. As $\alpha_Z$ increases relative to $\beta_Z/\alpha_Z$, fitness becomes constant over $\alpha_Z$. (C) The phenotype is a nonlinear function of the network steady state, $\beta_Z/\alpha_Z$, although largely linear around the phenotypic optimum. An optimum ratio at $\theta_{\beta_Z/\alpha_Z} = 0.8$ maximizes fitness. The dashed black line shows the phenotypic optimum. (D) The contributions of mutations in each molecular component to the adaptive walk. This is taken for populations with at least 2 steps in the walk. $\phi$ shows the difference in sums of absolute allelic effects among fixations in $\alpha_Z$ and $\beta_Z$ across an adaptive walk. Values greater than 0 represent larger changes in $\alpha_Z$ than $\beta_Z$, and vice versa for values smaller than 0. At 0, both $\alpha_Z$ and $\beta_Z$ changed by the same amount during the adaptive walk.

into Eq 1. We considered the case where $X = 1$, simplifying to

$$Z'(t) = -Z\alpha_z + \frac{\beta_Z}{2 + 2Z^8}$$

The equilibrium fulfills the equation

$$\frac{\beta_Z}{2\alpha_Z} = Z(1 + Z^8)$$

If we assume a model of simple gene regulation,

$$\theta = \frac{\beta_Z}{\alpha_Z}$$

$$\frac{\theta}{2} = Z(1 + Z^8)$$

where $\theta$ is the steady state of gene expression under simple gene regulation [52]. For small values of $Z$ (where $Z < 1$), this is well approximated by

$$Z = \frac{\theta}{2}$$

Using this, we approximate the solution of the ODE by the equilibrium whenever $X = 1$ and by 0 when $X = 0$. Hence, the solution can be given by

$$M = \int_{t_{start}}^{t_{stop}} \frac{\theta}{2} dt = \int_1^6 \frac{\theta}{2} dt = \frac{5\theta}{2}$$

When the optimum is at $P = M = 2$, the optimum steady state is

$$\theta_{\beta_Z/\alpha_Z} = \frac{2M}{5}$$
$$= 0.8$$

This assumes that the increase in $Z$ after $X$ is activated is rapid, and likewise for the decrease in $Z$ when $X$ is deactivated (i.e. in Fig 2B, the green curve should be close to rectangular around $X$ activation). This behavior is driven by $h$: for our chosen value, the above approximation predicts the observed $\theta_{\beta_Z/\alpha_Z}$ in Fig 6B.

To explore how the nonlinearity of the *GP* map might have influenced how populations navigated molecular component space, we examined the ratio of absolute additive effect sizes on molecular components among populations that adapted by two steps or more, denoted by $\phi$ (Fig 6D). Mathematically,

$$\phi = |2a_{\alpha_Z}| - |2a_{\beta_Z}|,$$

where $a$ is the allelic effect on $\alpha_Z$ or $\beta_Z$ without the multiplicative transformation (i.e. the value composed by Eq 6 instead of Eq 3. Allelic effects were multiplied by two since we simulated a diploid population. When $\phi > 0$, fixations have larger effects on the $\alpha_Z$ axis than the $\beta_Z$ axis, and vice versa for when $\phi < 0$. When $\phi = 0$, there are equal contributions from mutations affecting $\alpha_Z$ and $\beta_Z$. On average, $\phi = 0.164 \pm 0.150$, suggesting that $\alpha_Z$ mutations contributed most to adaptation, albeit by a small amount. Of the populations considered for this analysis, 29.5% adapted with only $\alpha_Z$ mutations, 25% with only $\beta_Z$ mutations, and the remaining 45.5% used a combination of both to reach the optimum.

## Discussion

### Overview

In this study, we explored how a simple gene network might influence trait evolution during an adaptive walk scenario. Models of adaptation have traditionally relied on direct genotype-phenotype relationships [42, 95, 96], overlooking the intricate gene interaction networks that drive trait expression at different stages of development (but see [68, 70, 71]). Understanding

how genetic networks influence adaptation is an under-explored area of research, particularly in the study of DFEs.

We observed that both network and additive models exhibit similarities in key aspects of their adaptive walks, specifically the number of adaptive steps and the distribution of fitness effects among both fixations and beneficial mutations (Figs 3 and 4). However, the characterization of the DFE among beneficial mutations was sensitive to the estimation method employed. Utilizing a pooling approach under the assumptions of the Central Limit Theorem leads to a homogenization of the shape parameter $\kappa$, nudging the distribution towards the Gumbel domain, and thus suggesting exponential spacing between adjacently-ranked beneficial alleles [17]. The averaging effect of pooling samples can mask underlying Fréchet or Weibull behaviors in $\kappa$, effectively reducing our ability to discriminate between different extreme value distributions.

In contrast, using a non-pooled approach preserves the contributions of each replicate's unique combination of mutations to the shape parameter, improving the detection of non-Gumbel behavior. However, this is not always possible in experimental studies where the number of beneficial mutations obtained per round of adaptation is small. Under these conditions, pooling between replicates is required to reduce the risk of incorrectly rejecting a Gumbel distribution [17, 88]. We suggest that researchers should be cautious of the homogenization effect of the pooling approach when estimating the DFE among beneficial mutations. Given these considerations, we next discuss the variation in estimates of the shape parameter $\kappa$ when using pooled and non-pooled approaches.

When we fit GPDs to each replicate with their own set of mutations, almost all the fitted GPDs belonged to the Weibull domain (S10 Fig). In addition, $\kappa$ was quite variable between simulations, models, and adaptive steps. $\kappa$ affects the GPD by modulating an upper bound on the effect size of beneficial mutations. With decreasing $\kappa$, this upper bound approaches zero. $\kappa$ was relatively stable in both models, but tended to decrease over longer walks under the network model and increase over longer walks in the additive model (particularly when comparing $\kappa$ before the optimum shift and at adaptive steps $\geq 3$ (S11 Fig and S2 Table). Regardless of this change, both models were largely Weibull-distributed at all adaptive steps (over 99% of simulations/adaptive steps belonged to the Weibull domain), which rejects Gillespie's Gumbel expectation.

Weibull-distributed DFEs of beneficial effects have been observed in a number of empirical studies in viral and bacterial populations (e.g. [16, 20, 29]). Furthermore, a Weibull distribution of effects is predicted when populations are close to an optimum and/or under stabilizing selection [18]. Hence, our results match theoretical expectations. Evidence for limits on adaptation seems common and both models seem to be affected by such limitations. While both models were starved of large-effect beneficial mutations relative to the exponential expectation, this effect was more limiting for network models than additive as the walk went on. This was further highlighted by a major difference between network and additive responses to selection: the rate of trait evolution.

## Trait adaptation and network evolution

In our simulations, we found that network simulations approached the optimum more slowly than additive models on average (Fig 3A). This difference in rate might not have only been due to the NAR structure, but also because of the difference in allelic effect scaling between additive and network models, as shown in Eqs 3 and 6. This scaling difference can influence rates of adaptation, mandating caution in attributing observed differences solely to the network's complexity. Hence, we do not make conclusions on how much the observed differences

are due to the NAR network's structure compared to the difference in allelic effect scaling. Nonetheless, there was a difference in the rate of adaptation between the models and the NAR DFE does not completely reflect the expectation under a multiplicative model, so the distribution of effects is at least partly mediated by the NAR structure. This will be the subject of a future study.

We explored the distribution of new mutations and discovered a complex DFE in the network models, contrasting with the simple distributions observed in additive models (Fig 5). The additive model distribution largely met Gillespie's expectations: the DFE was comprised of mostly neutral mutations, with a small number of beneficial mutations and a long tail of deleterious mutations (Fig 5, [97] pg. 267, Fig 6.5).

In the network model, we found a bimodal distribution of fitness effects. The bimodality is more pronounced than expected compared to a multiplicative model, suggesting that the NAR contributes to this shape. This observation aligns with previous studies that have reported complex and multimodal distributions of deleterious fitness effects (e.g. [98, 99]), potentially attributed to different classes of mutations with distinct DFEs [24]. A recent empirical study in *Escherichia coli* also found a similar bimodal DFE following a similar evolutionary regime [100]. Mutations in gene regulatory regions can have widely varying effects on fitness (e.g. [101–103]), suggesting that network structure can strongly affect the fitness distribution of mutants on a given molecular component [104]. The ridge-like shape of the $\alpha_Z/\beta_Z$ fitness landscape reflects this, showing that dependencies between the molecular components drive the fitness of organisms in the NAR model and that both the phenotypic and fitness effect of a mutation depends on its genetic background (Fig 6).

The additive model has a simple, one-dimensional fitness landscape: it recapitulates the fitness function, a normal distribution (Eq 5). This simplicity is not the case in the NAR model. The fitness landscape between the molecular components of the NAR revealed non-interchangeable per-locus effects (Fig 6A and 6C). The ridge-like shape of the landscape corroborates findings by Kozuch et al. [64], who investigated a similar NAR motif in the *E. coli lexA* transcription factor. NAR populations also faced lower beneficial mutation rates and a slight bias favoring $\alpha_Z$ mutations over $\beta_Z$, which might have been caused by the ridge-like landscape (Figs 5 and 6D). The bias suggests $\alpha_Z$ mutations should be less deleterious on average than $\beta_Z$ mutations. Supporting this, $\alpha_Z$ mutations were slightly less likely to be strongly deleterious, instead falling between the distribution's modes (S13 Fig). However, both mutation types remained largely deleterious, and had similar DFEs among beneficial mutations. A potential explanation for the bias towards $\alpha_Z$ mutations involves the optimal $\beta_Z/\alpha_Z$ ratio of 0.8 post-shift, versus 0.4 at burn-in. To reach this optimum from burn-in, populations could increase $\beta_Z$ or decrease $\alpha_Z$. However, the required $\alpha_Z$ decrease exceeded the $\beta_Z$ increase. In essence, larger $\alpha_Z$ mutations were needed to reach the new optimum. Crucially, $\alpha_Z$ and $\beta_Z$ mutations did not affect the phenotype equally.

## Complexity and the cost of adaptation

In our study, the NAR model demonstrated greater complexity compared to traditional additive models. We discovered a nonlinear relationship between $\beta_Z/\alpha_Z$ and phenotype, embodied by a nonlinear genotype-phenotype (GP) map (Fig 6C). Such maps are known to impose adaptive constraints [105, 106], reflected in the multimodal shape of the deleterious DFE for NAR populations (Fig 5A). The NAR model's increased complexity through multiple molecular components and dynamic biochemical interactions renders it more susceptible to deleterious mutations. This vulnerability could amplify Hill-Robertson interference, with beneficial alleles overshadowed by deleterious backgrounds [107]. For instance, consider the persistence of

epistasis over time. Gene interactions can lead to situations where genetic combinations have greater effects on fitness than the sum of their parts (positive epistasis), or the combinations can reduce fitness compared to the sum expectation (negative epistasis). Drift and mutation generate positive and negative epistasis in similar proportions [108]. However, when recombination is rare, selection will quickly fix synergistic combinations, leading to an excess of deleterious gene combinations which persist through drift and limit the efficiency of selection [107]. When a gene network introduces further epistasis on traits, this might compound the imbalance between negative and positive epistasis. Thus, network complexity may trade off with adaptive potential.

Trade-offs between complexity and adaptability echo the cost of complexity described by Orr [109]. The cost of complexity postulates that an increasing number of traits under selection leads to an increasingly deleterious mutation space, impeding adaptation. This cost could also apply to network models of traits, where molecular components under selection limit adaptation. While the NAR motif is simple, we were surprised by the complexity already apparent in the DFE. If even simple networks can generate such complexity, the cost of complexity might be more important for driving the evolution of traits than previously thought. Some empirical examples of these costs in network-mediated traits exist. Costanzo et al. [110] found that increasing the degree of genetic interaction in *Saccharomyces cerevisiae* genes was negatively correlated with single mutant fitness, suggesting an adaptive cost to maintaining a highly connected gene network. This cost might lead to the long-term stability of gene networks or complex traits: comparisons between *S. cerevisiae* and *Schizosaccharomyces pombe* genetic interaction networks revealed remarkable similarity in network structure [111]. Another study by Barua and Mikheyev [112] found features of housekeeping gene networks involved in reptile venom production were conserved between amniote clades, with those genes contributing to saliva production in mammals [112]. The cost of complexity suggests rather strong limitations on the evolution of complex genetic networks: in Orr's [109] model, the rate of adaptation declines according to the inverse square root of the number of traits, $1/\sqrt{n}$. Empirical evidence suggests that biological networks can be extremely complicated, connecting hundreds of nodes. Adaptation involving such a highly connected network would be extremely slow if Orr's [109] theory is correct. However, aspects of network structure, including local connectivity and modularity, might alleviate such a cost of molecular complexity.

One empirical example of network structure modulating the rate of evolution is the biofilm production network in *Candida* species [113]. In this network, seven master regulators control the expression of about one-sixth of the *C. albicans* genome and are required to drive biofilm production [113]. Genes that are connected to one or more of these master regulators are much less connected than the master regulators themselves and more "free" to evolve without affecting the expression of hundreds of other genes. Mancera et al. [113] found that the master regulators evolve slowly compared to the target genes, possibly due to the structure of the network. Since the master regulators affect the expression of hundreds of genes, the space of beneficial mutations at master regulator loci might scale according to the cost of complexity. On the other hand, the authors found that target genes are considerably more divergent between *Candida* species and populations, perhaps due to target genes harboring fewer interactions within the network than master regulators [113].

Despite the NAR motif being a simple network, we were surprised to find the adaptive constraints it appears to impose on populations. This point is underlined by the density of strongly deleterious alleles in the DFE of new mutations (Fig 5). Fixing multiple NAR molecular components (e.g. $K_Z$, $K_{XZ}$, $h$) renders the model's behavior more predictable. For sufficiently large

$\alpha_Z$, individual fitness correlates solely with the ratio $\beta_Z/\alpha_Z$ (Fig 6B), reflecting the steady-state cellular concentration under simple regulation [52]. This outcome may be attributed to our parameter choices; specifically, the rapid onset of $Z$ activation/suppression due to $h = 8$. Varying these Hill coefficients or treating them as evolvable components in future work could offer deeper insights into how more gradual responses to $X$ activation/deactivation and $Z$ production might modulate total $Z$ expression [52]. Splitting the Hill coefficient into an activation coefficient ($h_X$) and a repression coefficient ($h_Z$) would give further insight into the relative importance of $X$ and $Z$ responses to phenotype variation. Moreover, allowing these parameters to mutate could elucidate the full NAR fitness landscape and the cost of complexity apparent in a full NAR system.

Should $K_Z$ be allowed to evolve, the fitness landscape would likely exhibit increased ruggedness. In this context, the optimal $\beta_Z/\alpha_Z$ ratio would be contingent upon the $K_Z$ value. A similar effect is anticipated for $K_{XZ}$, the regulator of $Z$'s production rate in response to $X$. Consequently, both steady-state and fitness would depend on $K_Z$ and $K_{XZ}$. Introducing more evolvable molecular components is expected to amplify the landscape's ruggedness due to interdependencies that influence fitness, exacerbating the previously noted cost of complexity. To substantiate these projections, future research should explore more intricate networks, as exemplified by empirical studies such as Bertheloot et al. [114]. In addition, exploring the behavior of other network motifs will elucidate the generality of our findings.

There are other motifs that are common in nature that could be reasonable choices for a toy model of network-mediated traits. Other options include feedback loops, feed-forward loops, single input modules, and cascades. Each of these motifs have different behaviors that could lead to different evolutionary dynamics. For instance, feedback loops can create oscillatory expression patterns [115]. Autoregulation is a simple example of a feedback loop, however multi-locus feedback loops also exist. The p53–Mdm2 feedback loop is an example of a two-gene feedback loop in vertebrates. p53 is an important tumor suppressor transcription factor. Its expression is kept at an equilibrium cell concentration by the presence of Mdm2 [116]. p53 is a positive regulator of Mdm2, whilst Mdm2 inhibits p53 expression. When this balance is interrupted, increased Mdm2 production results in cell proliferation, whilst increased p53 production leads to cessation of growth [116]. Another common motif, the feed-forward loop, has eight different configurations, each with different behaviors [52]. Some of these configurations are more common in transcription networks than others, suggesting a selective pressure favoring these ubiquitous forms [117]. One of these common forms, the type I coherent feed-forward loop, is implicated in compensatory evolution via the emergence of epistasis [118].

## Ramifications of strong selection weak mutation (SSWM) and alternative approaches

This study focused on adaptation by employing a Gillespie-Orr model, which operates under the assumption of SSWM, characterized by $Ns \gg 1$ and $N\mu \ll 1$. We ensured adherence to the SSWM assumption by choosing an appropriate population size and mutation rate (see Table 1). In our simulations, $N\mu = 0.092$. However, $Ns$ varies due to selection coefficients being dependent on individuals' genetic backgrounds and phenotypes. Nonetheless, $Ns > 1$ holds at $s > 0.0002$ for this model, so most beneficial mutations should meet the $Ns \gg 1$ criterion. In the instance they do not, these mutations are driven by drift [119]. This also applies to slightly deleterious mutations. $N|s| = 1$ represents a "drift barrier", where the fitness effects of alleles become effectively too small to drive allele frequency changes, leading to drift-dominated dynamics.

The drift barrier explains our finding that the DFE among fixations followed a gamma distribution as opposed to an exponential in both network and additive models (S6 Fig). As populations approach the optimum, selection's efficiency is weakened by the drift barrier [119]. At the barrier, the probability of fixation ($\pi$) is no longer predicated by $\pi \approx 2s$ when the population is close to the optimum. Instead, $\pi = 1/2N$ becomes more applicable [120]. The switch from selection-dominated to drift-dominated dynamics occurs when $N|s| = 1$ [121]. In our simulations, this means that when $|s| < 0.0002$, drift is expected to dominate. Owing to the relatively large population size in our simulations ($N = 5000$), drift is unlikely to drive allele fixations early in the adaptive walk (i.e. when most beneficial alleles have $s > 0.0002$), however later steps might be more affected as the possible fitness improvement of any given allele faces diminishing returns.

Another assumption of the SSWM model is that only one mutation segregates at any given time point, i.e. before a new beneficial mutation arises, the previous beneficial mutation must have fixed. This was met most of the time (see S8 and S9 Figs), however, some simulations revealed instances where segregating variation contributed to adaptation, including the presence of alleles under balancing selection (S12 Fig). Future research should therefore explore the model using a polygenic approach, where many loci contribute small amounts to the molecular components underlying complex traits. Adaptation would then occur by small frequency shifts at these loci [122]. The genetic architecture of polygenic traits, that is, the number of loci, frequencies, and the distributions of their additive and non-additive effects on phenotypes, can greatly affect the response to selection [85]. There are also expectations for how genetic networks might determine the genetic architectures of polygenic traits: large-effect loci are likely linked to *cis*-regulatory features and coding region mutations, whilst small-effect loci are more likely to represent *trans*-regulatory elements [2, 123]. However, the non-additive effects of genetic networks on phenotypes remain understudied. These interactions have ramifications for theories of the evolution of recombination [124], and the persistence of linkage disequilibrium in natural populations [108].

A final point on SSWM is that it assumes that the fitnesses of a wild-type genotype and its mutants are uncorrelated [125]. This may not be the case: Cowperthwaite et al. [125] found that the fitnesses of simulated RNA mutations were correlated with those mutations' wild-type genotypes, violating the Gillespie-Orr model's assumptions. When mutations are correlated with the wildtype, the adaptive walk might be elongated, as the fitness benefit of a mutation is reduced relative to an uncorrelated beneficial allele [126]. However, work by Orr [127] showed that most results from the Gillespie-Orr model are robust to such correlations, especially those pertaining to the first step of the walk.

## Opportunities and challenges in modeling complex genetic networks

Our study serves as an introductory venture into the evolutionary modeling of genetic networks, elucidating how they guide evolutionary trajectories. The strong selection and weak mutation assumptions improve computational efficiency and interpretability but sideline rich polygenic dynamics [36, 85]. Our simplified NAR motif offers initial insights but lacks empirical complexity, making future work with more complex architectures important. Moving forward, our focus will shift towards a spectrum of genetic architectures to better capture non-additive effects. This will include investigating other network motifs, including several feed-forward loops. By doing so, we will enrich our understanding of how topology influences evolutionary pathways [128]. This transition will require further statistical and computational innovations to maintain feasibility. Collaborations with molecular biologists will enrich our models with further mechanistic details. Rigorous validation against empirical data and

existing theories will be a cornerstone in constructing more realistic models of adaptation (e.g. [114]).

To drive these future works, we have constructed a general form of our network model in S1 Appendix. In the full model, we envision a phenotype as a combination of "molecular traits": intermediate traits like methylation or gene expression which represent cellular or developmental processes [129]. Our NAR model represents the simplest case, where there is one molecular trait, $Z$ expression, which we treat as the phenotype. In more complex implementations, phenotypes can emerge from a composite of molecular traits. Each molecular trait has its own differential equation and its own set of molecular components driving gene expression, creating a hierarchical model of gene expression driving quantitative trait variation.

As we delve into more complex systems, computational hurdles become prominent. The complexity of the ODE system significantly impacts *in silico* performance [130, 131]. We are considering optimization strategies such as smarter caching of common genotypes, data-oriented design, and adaptive step-size algorithms. Pre-computed ODE solutions stored on disk are also under exploration as a means to tackle the computational load. Another option is to approximate ODE solutions via smooth functions or a deep neural network trained on previous simulated data. In summary, our study, while preliminary, opens up avenues for understanding the complex interplay between genetic networks and evolutionary dynamics, leaving us optimistic about the future of this research area.

## Conclusion

Our network approach contributes methodologically to the field and holds broad applicability to many open questions in evolutionary biology, from quantitative genetics to the mechanisms underpinning adaptability. It offers a robust framework to investigate issues such as the maintenance of genetic variation, the evolution of recombination, and the dynamics of adaptation from both standing variation and *de novo* mutations. We have introduced how networks can generate epistasis, with implications for the evolution of recombination, maintenance of linkage, and the trade-offs between network complexity and adaptability. Our study here shows that networks can create complex distributions of fitness effects that differ from additive expectations. While promising, it is essential to acknowledge the model's limitations, particularly concerning its computational demands and simplifications, which future work should aim to address. Empirical validation through evolutionary experiments will be crucial in assessing our model's real-world applicability. Expanding into adaptation by standing variation will further illuminate the role of genetic interactions during polygenic adaptation. As detailed systems models of complex traits become increasingly available, our framework can be employed to simulate the evolution of traits based on empirically-described networks. The utility of our model also extends beyond evolutionary biology, offering valuable insights for interdisciplinary collaborations that aim to integrate molecular biology, computational science, and statistical genetics. Future work should focus on leveraging this interdisciplinary potential and on exploring more complex genetic networks to better understand how gene interactions constrain evolution.

## Supporting information

**S1 Appendix. A hierarchical molecular network model for quantitative traits.** An extension of the negative autoregulation model to a general form to model quantitative traits under the control of an arbitrary genetic network.
(PDF)

**S1 Table. NAR model parameters.** Table of symbols, names, descriptions, and values for relevant parameters used in the NAR model.
(PDF)

**S2 Table. Generalized Pareto distribution parameter estimates.** Mean generalized Pareto distribution (GPD) parameters fit to mutant screen distributions of fitness effects over adaptive walks in additive and network populations. Brackets indicate 95% confidence intervals. Parameters were estimated by fitting a GPD to a random sample of mutations from the mutant screen experiments conducted on each replicate at each adaptive step. $\kappa$ is the shape parameter of the GPD. $\bar{\kappa}$ is the mean $\kappa$ value across $n$ replicate simulations. The log-likelihood ratio tests the null hypothesis that a sample of beneficial mutations belongs to an exponential distribution (which meets $\kappa = 0$). P-values across the $n$ replicates were combined using Fisher's method.
(PDF)

**S1 Fig. Flowchart of SLiM simulation procedure.** SLiM simulates a Wright-Fisher process to model evolutionary change through a genotype-phenotype-fitness (GPW) map. Our simulation began with 50,000 generations of burn-in to ensure populations were adapted to the environment. We then shifted the phenotypic optimum and adaptation was tracked for a further 10,000 generations. This process was repeated 2,880 times per model for replication purposes (i.e. there were 2,880 replicates of network and additive adaptive walks for a total 5,760 simulations). The *GPW* map consisted of several stages: first, the genotype was translated to phenotype via summing genetic effects at QTLs (for additive models) or by solving a system of ordinary differential equations (network models, *P/t* figure shows solutions to the differential equation for the blue and purple genotypes). Phenotype was then translated to fitness by a stabilizing selection fitness function (shown by the *w/P* figure—the purple phenotype has lower fitness than the blue phenotype). The fitness value then influenced the chance that an individual was sampled as a parent for the next generation. After parents were chosen, random mutations ($\mu$) could occur to introduce further genotypic and phenotypic variation in the next generation (shown in orange).
(TIFF)

**S2 Fig. Flowchart describing how selection coefficients and the distribution of fitness effects (DFE) among new mutations were estimated.** (A) Consider two alleles contributing to the phenotype (blue and purple boxes) at either the same or different loci. Individuals with both alleles have some phenotype ($P_1$), which gives rise to a fitness ($w_1$). By removing the purple allele and recalculating the phenotype, we achieve a different phenotype ($P_2$) and fitness ($w_2$). The difference between $w_1$ and $w_2$ represents the selection coefficient ($s$) of the purple allele. This difference can be measured either through addition—adding the purple allele to the genotype, or by knockout (removing the purple allele from the genotype). (B) To estimate the distribution of fitness effects (DFE) among new mutations, we conducted a mutation screen experiment. We generated mutants by taking 1,000 samples ($\epsilon$) from a standard normal distribution and adding those to the molecular component values from the SLiM simulations ($\alpha_Z$ and $\beta_Z$). Mutant phenotypes ($\mathbf{P_m}$) were calculated by inputting the mutant component values into an ordinary differential equation and solving it. The 1,000 samples were independently added to each molecular component to measure the DFE of both components. We then calculated the selection coefficients of $\epsilon$ by the addition method in (A). $P_m$ represents $P_1$ in (A), whilst the phenotypes without the added $\epsilon$ represents $P_2$. We then plotted the joint distribution of $s$ across all sampled $\epsilon$.
(TIFF)

**S3 Fig. Flowchart describing how waiting times to new beneficial mutations were calculated.** (A) For each of the 2,880 replicates, we took the distribution of $s$ calculated during the mutation screen experiments and extracted the proportion of new mutations with $s > 0$, $p_{s>0}$. This is the area under the curve shown in dark blue/red (to the right of the dashed line). This was repeated for each adaptive step ($N_{steps}$ within a replicate (i.e. replicates with more than one adaptive step had $p_{s>0}$ calculated for each adaptive step). The waiting time to a new beneficial mutation for a given replicate at a given step was calculated as $t_{wait} = 1/(2N\mu L_Q p_{s>0})$, where $N$ is the population size (multiplied by 2 because we simulate a diploid organism), $\mu$ is the per-locus, per-generation mutation rate, and $L_Q$ is the number of causal loci. (B) To calculate the difference between models in the waiting time to a new beneficial mutation, we ran a bootstrap analysis. We sampled 100,000 random pairs of additive and network models calculating $t_{wait}$ for each. We calculated the difference between their waiting times, $\Delta t_{wait}$. The random sampling gave a distribution of differences between network and additive waiting times. We used a paired t-test to determine if the mean difference in waiting times between models, $\bar{\Delta t}_{wait}$ was not zero (and hence there was a difference between models).
(TIFF)

**S4 Fig. Flowchart describing the estimation procedure for the shape of the distribution of fitness effects (DFE) among beneficial mutations via two approaches.** (A) Using the pooled distribution of $s$ across all simulations and adaptive steps that we calculated during the mutation screen experiments, we sampled 1,000 beneficial mutations (where $s > 0$) to generate a distribution of beneficial mutations. We then fit this to a generalized Pareto distribution (GPD) using the method outlined by Beisel et al. [88]. From the fit, we extracted the shape parameter of the fit, $\kappa$. This process was repeated 10,000 times to generate a distribution of $\kappa$. We treated the mean $\kappa$, $\bar{\kappa}$ as the estimate for the shape parameter. (B) We sampled $n_{x,y}$ mutations from each simulation's $s$ distribution at each adaptive step. The $s$ distributions were generated during the mutation screen experiments. $n_{x,y}$ was $\min(n_{s>0}^{x,y}, 100)$, where $n_{s>0}^{x,y}$ was the number of mutant screen alleles with $s > 0$ in replicate $x$ and adaptive step $y$. The sampled distributions were fit to a GPD as per (A), generating a distribution of $\kappa$ from each replicate and adaptive step. The mean was again treated as the estimate for the shape parameter.
(TIFF)

**S5 Fig. Distribution of generations at which mutations fixed in adapting additive (red) and network (blue) populations at each adaptive step.** There was no significant difference between populations at adaptive step 1. Generations are given relative to the optimum shift so that Generation 0 is when the optimum shifted.
(TIFF)

**S6 Fig. Zoomed view of Fig 4B.** The mode of the distribution is clearly greater than 0, possibly due to small effect alleles being more susceptible to loss by drift.
(TIFF)

**S7 Fig. Behavior of deleterious fixations in additive (red) and network (blue) models.** (A) The selection coefficient, $s$, decreases after the optimum shift at generation 50,000. (B) The allele frequency, $p$, of the fixations at the optimum shift.
(TIFF)

**S8 Fig. Mean observed heterozygosity in additive (red) and network (blue) populations over the adaptation period.** The mean is taken over 2,880 replicates per group. Heterozygosity was measured at the two QTLs (mQTLs in NAR models) in both models.
(TIFF)

**S9 Fig. Ratio of fixed effect phenotypes to mean population phenotype in additive and network models.** Values > 1 indicate that segregating variation decreases the mean population phenotype, and vice-versa for values < 1. Values = 1 indicate no segregating variation contributing to trait variance in the population.
(TIFF)

**S10 Fig. Comparison of sampled generalized Pareto distribution (GPD) fits to the distributions of fitness effects (DFE) among beneficial mutations fit onto mutations sampled by two methods.** Mutations were created by a mutant screen experiment where 1,000 mutations were randomly generated and their fitness effects measured relative to the phenotype of a population at a given adaptive step. The pooled method (A) pooled together the mutations from all mutant screens across simulation replicates and adaptive steps and used bootstrapping to repeatedly sample 1,000 alleles from the pooled distribution to fit a GPD onto. The non-pooled method (B) fit the GPD onto a sample of 100 mutant screen alleles from each replicate simulation at each of its adaptive steps. Each figure shows the resulting GPD fits of 5 randomly sampled replicates per model of these two methods (i.e. either 5 simulations in the non-pooled case or 5 repeated samples of 1,000 alleles for the pooled case). Note that fewer than 5 samples existed for adaptive steps > 1 in (B), owing to the rarity of walks that long.
(TIFF)

**S11 Fig. Comparison of mean shape parameters, $\kappa$, from generalized Pareto distribution (GPD) fits to beneficial alleles randomly sampled from each simulation's mutant screen experiment between models and at each adaptive step.** $\kappa$ describes the shape of the GPD, with negative values (seen here) indicating a Weibull domain of attraction. As $\kappa$ decreases, the maximum size of a beneficial mutation decreases. Error bars are 95% confidence intervals. Sample sizes for each group are given in S2 Table.
(TIFF)

**S12 Fig. Two examples of alleles under balancing selection during adaptation in both additive (red) and network (blue) models.** The additive mutation had a phenotypic effect $\alpha = -0.618$ while the NAR allele had a phenotypic effect $\alpha = -0.689$.
(TIFF)

**S13 Fig. The distribution of fitness effects among new mutations in network models for both $\alpha_Z$ and $\beta_Z$ molecular components.** Compared to $\beta_Z$ mutations, $\alpha_Z$ mutants were less likely to have strongly deleterious effects and more likely to have slightly deleterious effects existing on one of the two modes of the distribution. The shape of the distribution was similar between the molecular components.
(TIFF)

## Acknowledgments

We would like to acknowledge members of the Ortiz-Barrientos lab for their feedback on earlier versions of this manuscript. This research was conducted using the Gadi HPC system maintained by the National Computational Infrastructure (NCI), which is supported by the Australian Government.

## Author Contributions

**Conceptualization:** Jan Engelstädter, Daniel Ortiz-Barrientos.

**Data curation:** Nicholas L. V. O'Brien.

**Formal analysis:** Nicholas L. V. O'Brien.

**Funding acquisition:** Daniel Ortiz-Barrientos.

**Investigation:** Nicholas L. V. O'Brien.

**Methodology:** Nicholas L. V. O'Brien, Barbara Holland, Jan Engelstädter, Daniel Ortiz-Barrientos.

**Project administration:** Nicholas L. V. O'Brien.

**Resources:** Nicholas L. V. O'Brien.

**Software:** Nicholas L. V. O'Brien.

**Supervision:** Barbara Holland, Jan Engelstädter, Daniel Ortiz-Barrientos.

**Validation:** Nicholas L. V. O'Brien.

**Visualization:** Nicholas L. V. O'Brien.

**Writing – original draft:** Nicholas L. V. O'Brien.

**Writing – review & editing:** Nicholas L. V. O'Brien, Barbara Holland, Jan Engelstädter, Daniel Ortiz-Barrientos.

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
