## [Decision Letter · Decision Letter 0]

24 Mar 2024

Dear Dr O'Brien,

Thank you very much for submitting your Research Article entitled 'The distribution of fitness effects during adaptive walks using a simple genetic network' to PLOS Genetics.

The manuscript was fully evaluated at the editorial level and by independent peer reviewers. The reviewers appreciated the attention to an important topic but identified some concerns that we ask you address in a revised manuscript.

We therefore ask you to modify the manuscript according to the review recommendations. Your revisions should address the specific points made by each reviewer.

Yours sincerely,

Robert L. Unckless, Ph.D.

Guest Editor

PLOS Genetics

Kelly Dyer

Section Editor

PLOS Genetics

Three reviewers (and I) have read your manuscript entitled "The distribution of fitness effects during adaptive walks using a simple genetic network" and all reviews were largely positive. This detailed simulation study appears to be very carefully conducted and contains several interesting findings that will be of interest to geneticists studying adaptation and gene regulatory networks. Several reviewer points should be addressed in a minor revision. One that I think is particularly important is working on readability for a broad audience. The subject matter is difficult, and overall, authors have done an excellent job of explaining the context, approach and results. Two specific suggestions are to reduce acronyms (these are distracting and similar enough that it is difficult to keep them all straight), and to reduce subscripts that are unnecessary (see R1's comments). It would also be useful to present the full set of differential equations prior to equation 1. R1 also asks for a justification of using SLIM when the SSWM model and simple network might allow for a simpler (and more computationally efficient) approach. R2 makes important points about the relevance of the cost of complexity discussion when the network has only 2 genes and is in the SSWM parameter space. R2 also notes that the dependence of the distribution of fitness effects on evolutionary history has limited relevance to the question - and this could be mentioned. R3 has several suggestions about comparisons and calculations of fitness landscapes.

Reviewer's Responses to Questions

**Comments to the Authors:**

Reviewer #1: O'Brien et al. use a simple gene network model to investigate the distribution of fitness effects (DFE) during the course of adaptive evolution. They model a quantitative trait under stabilizing selection determined by the negative autoregulation (NAR) network motif. They run evolutionary simulations to investigate adaptive walks following a shift to a new phenotypic optimum and use a simple additive model as a control. O'Brien et al. find that the NAR and additive models behave in a broadly similar manner but with some interesting differences. For example:

(a) Populations adapted more slowly under the NAR model than under the additive model.

(b) The DFEs of populations adapting under the NAR model had fewer beneficial mutations than those of populations adapting under the additive model.

The research is topical and is likely to appeal to a broad audience of geneticists (evolutionary, population, quantitative, systems, etc). The methods are sound. The paper is clearly written and well-argued.

I leave the authors with some minor comments/suggestions (in order of appearance).

1. In lines 41 and 46 you write that some distribution is "analogous" to another distribution? I don't understand what that means and to what end the analogies are being made.

2. Since your study was conducted in the SSWM regime (line 334), was there a reason not to use the more efficient approach of simulating an adaptive walk by sampling a candidate mutation and accepting or rejecting it as a substitution based on the probability of fixation? If so, the reason should be stated.

3. There are a lot of subscripts. The subscript in H_O (Eq. 7) seems unnecessary since no other kind of heterozygosity is used. Also, it is confusing because the subscript O is used to designate "optimal" in P_O (line 271).

Another subscript that may not be needed is "fs" in r_fs (Eq. 8). Also, the subscript is capitalized in the Results (e.g., line 462) but not in Eq. 8.

4. Are the effects of mutations described in lines 347-379 assayed in a heterozygous or homozygous state? Please clarify.

5. In line 786 you write: "There are other motifs that are common in nature that could be reasonable choices for a toy model of network-mediated traits. Other options include feedback loops, feedforward loops..." You appear to have missed a pioneering study of compensatory evolution using a model of the type I coherent feedforward loop network motif from a decade ago.

K. Bullaughey. Multidimensional adaptive evolution of a feed-forward network and the illusion of compensation. Evolution 67(1): 49-65, 2013.

I think you'll find it interesting.

6. In line 830 you mention that some alleles were observed to experience balancing selection. Presumably these were cases of overdominance. I know this is beyond the scope of this paper but I would be interested to see the distribution of dominance effects (see also my comment #4).

Reviewer #2: This paper appears to be a very carefully developed and produced computational study. I have only minor comments on what the paper presents. But there are issues with what the paper doesn't present that I will comment on. I will not be asking that any of the work in the submission be re-done, but only that the issues I raise be addressed in the manuscript.

The introduction to Gillespie-Orr theory is welcome. Many papers on the subject have been published. So the following issue is not unique to this submission. The assumption behind Gillespie-Orr theory is that one is taking i.i.d. samples from some probability distribution. But there is no mechanistic basis why that model should be appropriate for the mutational neighborhoods of organismal genotypes.

There is an important point never mentioned in the paper, which is that the dependence of the DFE on the evolutionary history enters ONLY as it sets the values of \\alpha_Z and \\beta_Z. In other words, \\alpha_Z and \\beta_Z screen-off (Salmon 1971) the prior history of the population. So all we need to find out about the DFE is just to calculate it over the the different points on the 2-D fitness landscape. To sample it only for the endpoints of evolutionary trajectories gives us only some snapshots of the DFE.

So question of whether or not the SSWM regime persists is irrelevant to the DFE except in how it affects the evolutionary trajectories of \\alpha_Z and \\beta_Z.

Mutation consists of i.i.d. perturbations of the \\alpha_Z and \\beta_Z values, which the dynamics map to fitnesses. Because the fitnesses are bounded above, if any extreme value distributions are applicable it would be the Weibull Type III.

With only two parameters evolving, it is hard to see how the results here have any bearing on the "cost of complexity" issue brought up many times throughout the paper.

p. 20 discusses the evolution of more complex gene networks. There is a substantial literature on the evolvability of gene networks that appears divorced from the current study. Indeed "evolvability" appears nowhere in the paper.

Minor Comments:

The paper never explicitly states that L_Q = 2 in all the simulations. While we learn that it is 2 genes on p. 4 line 127, nowhere is L_Q = 2 stated. Supplement Fig 1 is misleading in that it shows more than 2 genes but makes no mention that in the simulations, only 2 are considered.

p. 6 line 189: It is hard to parse K_{XZ}^{n_{XZ}}, etc. If you can find a way to make it more readable, that would be good. Also, K is never defined.

p. 6 line 213: "The constant values of the other..." That should be moved up to the first appearance of the symbols to indicate they are not varied.

p. 7 line 247. The first appearance of L_Q. Here is where it should be stated that all the simulations run in the paper assume L_Q = 2. If that is not correct, then I am really stymied as to what values L_Q takes in the work.

p. 8 line283: "per locus per generation" The reader should be reminded that there are only 2 loci in the simulations.

p. 8 line 310: "the optimum shift". This is the first we are told of this. It is so fundamental that it shouldn't be buried in the middle of a middle paragraph. There is a fundamental difference between populations near mutation-selection balance and those that are far from balance, so the presence of an optimum shift, and its magnitude, are core aspects of the model. It should be included in the Intro on p. 2.

p. 9 Table 1. It should include L_Q.

p. 9 The section describing H_O^i and r_{fs} never explain how they will be used to validate the SSWM conditions. Something should be stated here.

p. 10 line 348: "An important part of adaptive walk theory is the shape of the DFE." This is the whole purpose of the paper so it is a bit odd to call it "an important part".

p. 10 line 357: eq (9). In adaptation on a fitness landscape, what enters the dynamics is not the absolute difference in fitnesses, but the relative increase. So I would think a better definition is s = (w_k / w) - 1.

Fig. S10B: The curves all look like hypberbolas. Is that real or artifact or what? They are so sparse that I would not want to infer that they belong to any particular distribution type.

p. 21 line 847. "The strong selection and weak mutation assumptions grant computational tractability"

Since the paper uses individual-based simulations with the Wright-Fisher model, the most computationally intensive aspect is already included. There would be no added computational cost to shifting to strong mutation.

Supplemental figures: The figure labels should be on the graphic itself, not just the separate file titles. It is a bit annoying for reviewers to have to piece together the paper from the main file, supplement file, and supplemental figure files.

p. 22: The points about future computational savings are well taken. The authors already computed the adaptive landscape of the two variables \\alpha_Z and \\beta_Z. It should be possible to closely approximate that with smooth functions, or even a deep neural network, to emulate the ODEs, and save on computation.

Reviewer #3: Review is attached.

**Have all data underlying the figures and results presented in the manuscript been provided?**

Reviewer #1: Yes

Reviewer #2: Yes

Reviewer #3: Yes

PLOS authors have the option to publish the peer review history of their article (what does this mean?). If published, this will include your full peer review and a

---

## [Editor Report · Decision Letter 1]

4 May 2024

Dear Dr O'Brien,

We are pleased to inform you that your manuscript entitled "The distribution of fitness effects during adaptive walks using a simple genetic network" has been editorially accepted for publication in PLOS Genetics. Congratulations!

Yours sincerely,

Robert L. Unckless, Ph.D.

Academic Editor

PLOS Genetics

Kelly Dyer

Section Editor

PLOS Genetics

Comments from the reviewers (if applicable):

Thank you for your thoughtful revisions to the prior submission. The revisions make the piece easier to understand, especially for the non-specialist. I also appreciate the discussion of the complexity problem.

**Data Deposition**

http://datadryad.org/submit?journalID=pgenetics&manu=PGENETICS-D-24-00130R1

**Press Queries**

---

## [Editor Report · Acceptance letter]

21 May 2024

PGENETICS-D-24-00130R1 

The distribution of fitness effects during adaptive walks using a simple genetic network 

Dear Dr O'Brien, 

We are pleased to inform you that your manuscript entitled "The distribution of fitness effects during adaptive walks using a simple genetic network" has been formally accepted for publication in PLOS Genetics! Your manuscript is now with our production department and you will be notified of the publication date in due course.

With kind regards,

Anita Estes

PLOS Genetics

On behalf of:
